# TLR4 signalling via Piezo1 engages and enhances the macrophage mediated host response during bacterial infection

Jing Geng [1,2,5], Yiran Shi[1,5], Jinjia Zhang[1,5], Bingying Yang[1], Ping Wang[1], Weihong Yuan[2], Hao Zhao[1], Junhong Li[1], Funiu Qin[1], Lixin Hong[1], Changchuan Xie [1], Xianming Deng[1], Yujie Sun [3], Congying Wu [4], Lanfen Chen [1✉] & Dawang Zhou [1✉]

TLR4 signaling plays key roles in the innate immune response to microbial infection. Innate immune cells encounter different mechanical cues in both health and disease to adapt their behaviors. However, the impact of mechanical sensing signals on TLR4 signal-mediated innate immune response remains unclear. Here we show that TLR4 signalling augments macrophage bactericidal activity through the mechanical sensor Piezo1. Bacterial infection or LPS stimulation triggers assembly of the complex of Piezo1 and TLR4 to remodel F-actin organization and augment phagocytosis, mitochondrion-phagosomal ROS production and bacterial clearance and genetic deficiency of Piezo1 results in abrogation of these responses. Mechanistically, LPS stimulates TLR4 to induce Piezo1-mediated calcium influx and consequently activates CaMKII-Mst1/2-Rac axis for pathogen ingestion and killing. Inhibition of CaMKII or knockout of either Mst1/2 or Rac1 results in reduced macrophage bactericidal activity, phenocopying the Piezo1 deficiency. Thus, we conclude that TLR4 drives the innate immune response via Piezo1 providing critical insight for understanding macrophage mechanophysiology and the host response.

[1] State Key Laboratory of Cellular Stress Biology, Innovation Center for Cell Signaling Network, School of Life Sciences, Xiamen University, Xiamen, Fujian, China. [2] National and Local Joint Engineering Research Center of Biodiagnosis and Biotherapy, The Second Affiliated Hospital of Xi'an Jiaotong University, Xi'an, Shaanxi, China. [3] Biomedical Pioneering Innovation Center (BIOPIC), School of Life Sciences, Peking University, Beijing, China. [4] Institute of Systems Biomedicine, Peking University Health Science Center, Beijing, China. [5] These authors contributed equally: Jing Geng, Yiran Shi, Jinjia Zhang. ✉email: chenlanfen@xmu.edu.cn; dwzhou@xmu.edu.cn

Phagocytes are specialized immune cells that engulf harmful microorganisms and destroy them in phagosomes. Toll-like receptors (TLRs) on a phagocyte are essential for host defense against infection as they recognize invading microorganisms and initiate signaling pathways that launch innate immune responses and further instruct development of antigen-specific acquired immunity to destroy invaders[1–3]. Phagocyte destruction of engulfed microorganisms depends on the production of bactericidal reactive oxygen species (ROS) by both phagosomal NADPH oxidase machinery and the mitochondria translocated at the phagosomes[4,5]. Our previous study demonstrated that Hippo kinases Mst1 and Mst2 (Mst1/2) activated by TLR signaling are key regulators of phagocytosis and microbe-elicited ROS production[5]. Consistently, a loss-of-function mutation in human Mst1 resulted in recurrent bacterial infections, viral infections, mucocutaneous candidiasis, cutaneous warts, and skin abscesses[6,7]. Mst1 deficiency in mice exhibited similar phenotype indicating that Mst1 exerts a crucial role in innate immune defense against infection[8–13]. However, the activation mechanism of Mst1/2 by TLR for host defense remains elusive.

Recent studies showed that the alteration in the physical microenvironment such as extracellular matrix (ECM) stiffness and architecture becomes one of the hallmarks of tissue infection or cancer[14,15]. Thus, a change in the external force will naturally cause a change in the mechanical tension of the cell and eventually lead to biological activities. Mechanotransduction involves the sensing of mechanical cues by cells, the transduction of these cues into molecular signals, and the modulation of gene and protein expression and hence cellular functions. Mechanosensitive ion channels (MSICs), which sense external mechanical forces and translate them into intracellular events through their downstream effectors, are involved in the aforementioned processes[16,17]. Piezo1 and Piezo2, recently identified mechanically activated ion channels with high affinity for calcium, are evolutionally conserved and involved in development, differentiation, and growth of multiple tissues and activated by various types of mechanical stimuli in many cells[18–30]. When phagocytes probe and migrate through the surrounding matrix, they might sense mechanical cues to change their shape and behaviors[31–33]. Recent studies showed that Piezo1 modulates inflammatory response, indicating Piezo1 might play an important role in immunity[25,31,33–35].

Here we report that the TLR4 coordinates the mechanical sensor Piezo1 to activate CaMKII-Mst1/2 axis for coaxing macrophages into achieving the functions necessary for host defense. Piezo ion channels are generally believed to be activated by various types of mechanical stimuli and function as biological pressure sensors. Interestingly, we found that LPS, a TLR4 ligand, or bacterial infection triggers assembly of the complexes of Piezo1 and TLR4 to remodel F-actin organization and promote phagocytosis, mitochondrion–phagosome juxtaposition for ROS production and bacterial clearance, whereas knockout of Piezo1 impairs these responses. We further revealed that LPS induces TLR4 to activate Piezo1-mediated calcium influx, which turns on the CaMKII-Mst1/2-Rac axis to enhance macrophage bactericidal activity. Consistently, inhibition of CaMKII or knockout of either Mst1/2 or Rac1 results in reduced bactericidal activity, phenocopying Piezo1 deficiency. Thus, we conclude that Piezo1 governed mechanotransduction system is essential for TLR4-driven innate response against pathogen invasion.

## Results

**Piezo1 is associated with TLR4 upon bacterial infection.** To study the function of mechanosensitive ion channels (MSICs) in innate immunity, we determined the expression levels of MSICs on macrophages. The result showed that the mRNA levels of *Piezo1*, but not *Piezo2* or other reported mammalian MSICs, were highly expressed in macrophages (Supplementary Fig. 1a). The expression of Piezo1, but not Piezo2, in myeloid cells was confirmed by using *Piezo1*[P1tdT] mice, in which a Piezo1-tdTomato fusion protein is expressed[26] and *Piezo2*-GFP-IRES-Cre knock-in reporter mice, in which a GFP-tagged Piezo2 protein is expressed[23] (Fig. 1a and Supplementary Fig. 1b). To characterize the role of Piezo1 in innate immune cells upon infection, we prepared BMDMs from the *Piezo1*[P1tdT] transgenic mice and infected them with *E. coli*. Interestingly, we observed that the subcellular Piezo1, visualized by tagging the fluorescent protein tdTomato, was condensed and colocalized with TLR4 and *E. coli* in BMDMs (Fig. 1b). We then incubated BMDMs isolated from *Piezo1*[P1tdT] or TLR4 deficient *Piezo1*[P1tdT] (*Tlr4*[−/−]*Piezo1*[P1tdT]) mice with uncoated or LPS-coated latex beads to elicit phagocytosis of these beads. Similarly, higher amount of Piezo1 was recruited and colocalized with TLR4 on the LPS-coated beads than that on the uncoated beads, whereas no obvious Piezo1 or TLR4 could be detected on the beads engulfed in TLR4 deficient macrophages (Fig. 1c). Surprisingly, treatment of soluble LPS also resulted in the colocalization of Piezo1 and TLR4 receptor, as observed with a structure illumination microscopy (SIM) approach (Fig. 1d) suggesting that a soluble TLR4 ligand can also trigger the association of TLR4 with Piezo1. We then confirmed the association of TLR4 with Piezo1 by coimmunoprecipitation (Co-IP) (Fig. 1e) and fluorescence resonance energy transfer (FRET) assays (Fig. 1f). In addition, the immunoprecipitation of TLR4 in BMDMs treated with LPS, the Piezo1 agonist Yoda1 or LPS plus Yoda1 retrieved several endogenous proteins selectively, as identified by mass spectrometry, confirming the increase in the interaction of Piezo1 with TLR4 after LPS activation or Yoda1 treatment, and with the highest peptides number detected upon the stimulation of LPS plus Yoda1 (Fig. 1g and Supplementary Fig. 1c). Taken together, these results suggested that Piezo1 might coordinate TLRs signalling in response to bacterial infection.

**Piezo1 deficiency impairs bactericide of macrophage.** These observations promoted us to investigate whether and how Pizeo1 is involved in host defense against bacterial infection. We ablated Piezo1 in myeloid cells by crossing *Piezo1*[fl/fl] mice (LoxP-flanked *Piezo1* allele) with mice expressing Cre recombinase driven by the myeloid cell-specific promoter of the gene encoding lysozyme M (*Lyz2*-Cre). *Piezo1*[fl/fl] *Lyz2*-Cre mice and their *Piezo1*[fl/fl] wild-type littermates were born at the expected ratio and exhibited no substantial differences in the number of circulating lymphocytes, monocytes, and granulocytes, according to peripheral blood counts (Supplementary Fig. 2a). Flow cytometry indicated that the frequency of Gr-1[+]CD11b[+] neutrophils and F480[+]CD11b[+] macrophages in the bone marrow, spleen, and blood, and the composition and activation status of T cells and B cells in the spleen and lymph nodes were similar in *Piezo1*[fl/fl] and *Piezo1*[fl/fl] *Lyz2*-Cre mice (Supplementary Fig. 2b-d). We then constructed a model of septic peritonitis, cecal ligation, and puncture (CLP), on *Piezo1*[fl/fl] and *Piezo1*[fl/fl] Lyz2-Cre mice. The bacterial peritonitis killed ~80% of *Piezo1*[fl/fl] *Lyz2*-Cre mice but only ~40% of *Piezo1*[fl/fl] littermates (Fig. 2a), and the bacterial invasion of the spleen, lung, liver, and kidney after CLP induction was significantly higher in *Piezo1*[fl/fl] *Lyz2*-Cre mice than that in *Piezo1*[fl/fl] control littermates (Fig. 2b). These results suggested that mice lacking Piezo1 in innate immune cells were more susceptible to bacterial infection.

As a mechanically activated ion channel, Piezo1 plays an important role in transducing matrix mechanical cues to intracellular signaling. We then investigated the bacterial killing

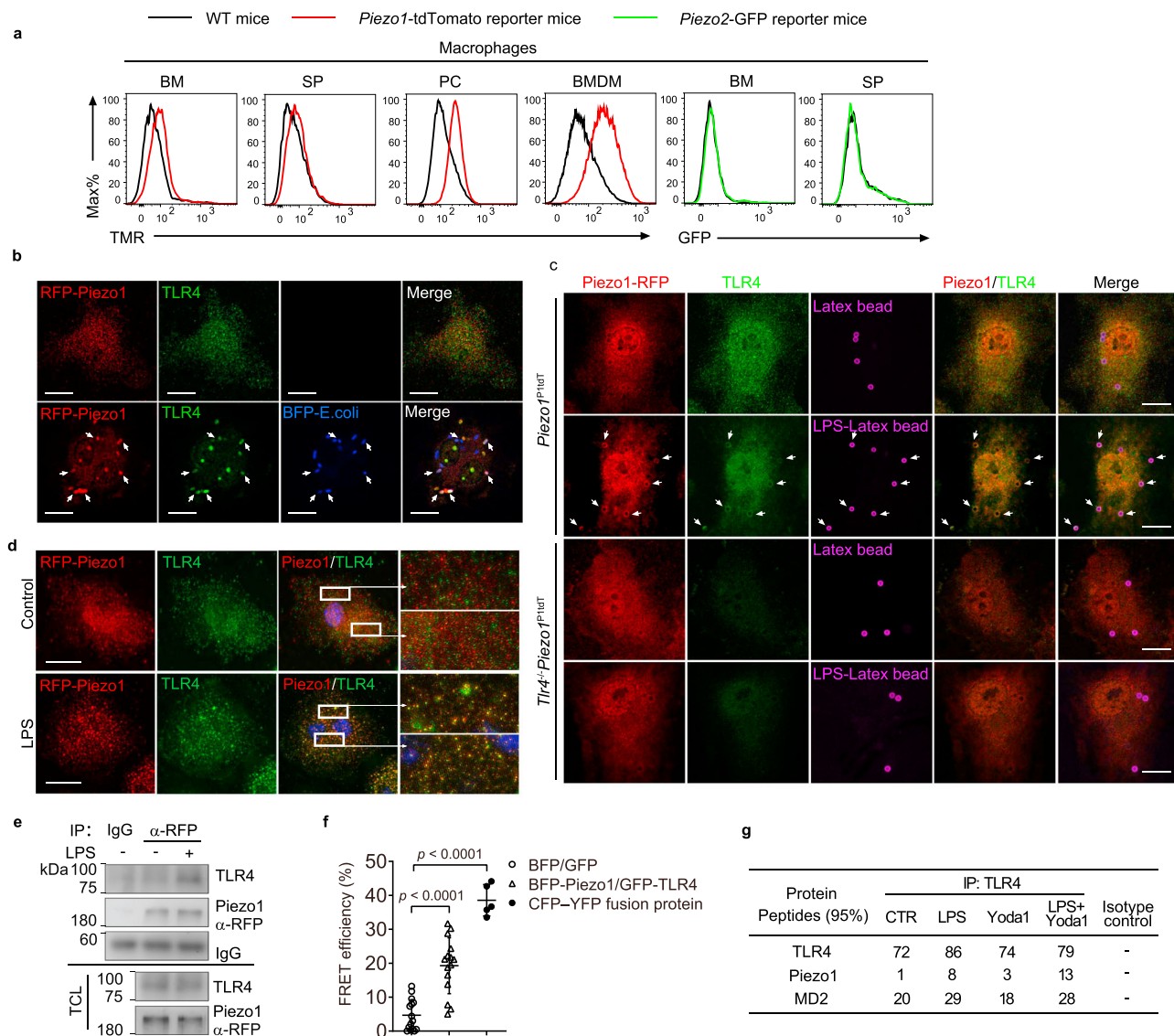

**Fig. 1 Piezo1 is induced and associated with TLR4 upon infection. a** Flow cytometry histograms of tdTomato red fluorescence (TMR) or GFP of macrophages (CD11b[+]F4/80[+]) from bone marrow (BM), spleen (SP), peritoneal cavity (PC), or bone marrow-derived macrophage (BMDM) of $Piezo1^{P1-tdT}$ mice or BM, SP of $Piezo2$-EGFP-IRES-Cre mice. **b** Confocal microscopy of the colocalization (white arrows) of Piezo1 (red), TLR4 (green), and $E.\ coli$ (blue) in Piezo1[P1tdT] BMDMs infected with $E.\ coli$ (MOI, 20). Scale bar = 20 μm. **c** Confocal microscopy of $Piezo1^{P1tdT}$ or $Tlr4^{-/-}Piezo1^{P1tdT}$ BMDMs incubated with uncoated or LPS-coated latex beads, followed by immunostaining of Piezo1 or TLR4 as indicated; white arrows indicate colocalization of beads (purple), TLR4 (green), and Piezo1 (red). Scale bars, 20 μm. **d** SIM of the colocalization of Piezo1 (red) and TLR4 (green) in $Piezo1^{P1tdT}$ BMDMs treated with or without LPS (1 μg ml$^{-1}$) for 30 min; ×16 magnification of areas outlined in the main images are shown next to the main images. Scale bars, 20 μm. **e** Immunoblot analysis of BMDMs expressing RFP-tagged Piezo1 treated with or without LPS, immunoprecipitated with anti-RFP or control IgG, and analyzed by immunoblot with anti-TLR4 or anti-RFP antibodies; below, immunoblot analysis of total cell lysates (TCL) without immunoprecipitation. Uncropped blots in Source data. **f** The FRET efficiency from BFP to GFP in the cells co-transfected with BFP-Piezo1 and GFP-TLR4 (n = 15 cells), CFP and YFP (n = 15 cells), or BFP–GFP fusion protein (n = 5 cells), respectively. Data are presented as mean +/− SD. The $p$-values of two-tailed unpaired Student's $t$ test are indicated. **g** Identification of TLR4, Piezo1, and MD2 by mass spectrometry in immunoprecipitation assays using TLR4 antibody in cell lysates of BMDMs untreated or treated with LPS (1 μg ml$^{-1}$), Yoda1 (5 μM), or both. One experiment representative of three independent experiments with similar results. Source data are provided as a Source data file.

activity of BMDMs by inoculating them on different stiffness culture matrix, 5 kPa, 35 kPa, or glass. Bacterial killing assays showed that much fewer phagocytosis events, i.e., the number of live intracellular bacteria at the 0 time-point, were found in BMDMs cultured on the soft matrix (5 kPa) than that on the stiffer matrix (Fig. 2c). In contrast, lower bactericidal activity or higher number of the remaining live intracellular bacteria at 90 and 120 min after $E.\ coli$ infection was found in BMDMs cultured on soft matrix than that of stiffer matrix (Fig. 2c). These results

indicated that the proper extracellular matrix stiffness is required for macrophage phagocytosis and bactericidal activity. Further experiments revealed that Piezo1 deficiency impaired bacterial engulfment and clearance in BMDMs cultured on glass or 35 kPa matrix, but not BMDMs cultured on 5 kPa matrix (Fig. 2d). In addition, flow cytometry analysis showed that Piezo1-deficient cells exhibited less phagocytosis of FITC-labeled $E.\ coli$ than did wild-type cells on regular culture plates (Fig. 2e). Furthermore, we observed that Yoda1 treatment can enhance bacterial

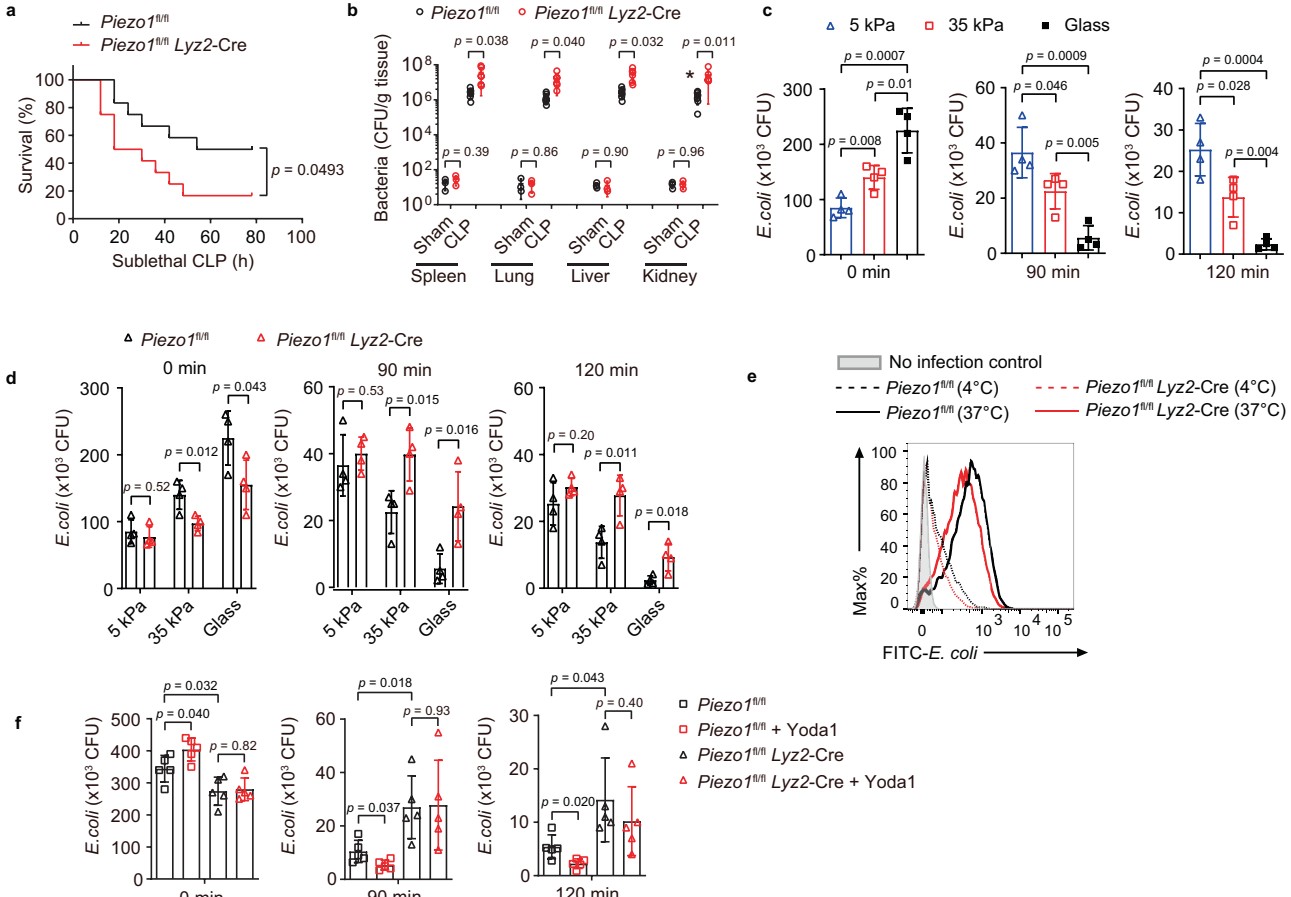

**Fig. 2 Piezo1 deficiency impairs phagocytosis and bactericide of macrophage. a**, **b** *Piezo1*fl/fl or *Piezo1*fl/fl *Lyz2*-Cre mice ($n = 12$ mice per group per experiment) subjected to sublethal CLP. The *p*-value of mortality Mantel–Cox test is indicated (**a**), and the bacterial loads (colony forming units, CFU) measured in the lung, liver, spleen, and kidney after CLP induction (Sham group $n = 3$, CLP group $n = 6$) (**b**). **c**, **d** Pathogen burden (CFU) in wild-type (**c**, **d**) or Piezo1-deficient (**d**) BMDMs infected with *E. coli* (MOI, 20) for 0, 90, or 120 min on different stiffness culture matrix, 5 kPa, 35 kPa, or glass ($n = 4$ independent samples). **e** Flow cytometry of *Piezo1*fl/fl or *Piezo1*fl/fl *Lyz2*-Cre BMDMs left uninfected or infected for 20 min at 37 or 4 °C with FITC-labeled *E. coli* (FITC-*E. coli*) at a MOI of 20. **f** Pathogen burden in *Piezo1*fl/fl and *Piezo1*fl/fl *Lyz2*-Cre BMDMs treated with or without Yoda1 (5 μM, 30 min), followed by *E. coli* infection (MOI, 20) for 0, 90, or 120 min ($n = 5$ independent samples). Data are presented as mean $+/-$ SD and the *p*-values of two-tailed unpaired Student's *t* test are indicated in (**b**), (**c**), (**d**), and (**f**). Data are from one experiment representative of three independent experiments with similar results. Source data are provided as a Source data file.

phagocytosis and clearance by macrophages, and this effect cannot be seen in Piezo1-deficient cells (Fig. 2f). These data suggested that matrix mechanical force and Piezo1 are critical for phagocytosis and efficient clearance of bacteria for host defense.

**Piezo1 regulates cytoskeleton remodeling and ROS production.** We next sought to investigate the mechanism by which Piezo1 regulates macrophage phagocytosis and bactericidal activity. The phagocytosis of particles by macrophages depends primarily on the reorganization of actin cytoskeleton. We observed that BMDMs formed filopodia containing tight bundles of long actin filaments covered with cell membrane upon Yoda1 stimulation (Fig. 3a). Interestingly, the enhanced filopodia formation is correlated with higher cellular stiffness measured with the atomic force microscopy (AFM), as shown by the increased average cell stiffness upon Yoda1 treatment, but the decreased average cell stiffness after Cytochalasin D (CytD) treatment that disrupts actin polymerization (Fig. 3a, b). Consistently, compared to wild-type BMDMs, Piezo1-deficient BMDMs did not form obvious filopodia or actin accumulation at sites of bacteria (Fig. 3c) and

exhibited the lower average cellular stiffness measured by AFM (Fig. 3d) under various stimulations, such as LPS, *E. coli*, and Yoda1, suggesting that Piezo1 is required for macrophage cytoskeletal reorganization, filopodia formation, and cellular stiffness during innate immune responses.

It has been previously shown that the cytoskeletal reorganization was also required for the juxtaposition of phagosomes containing intracellular pathogens and mitochondria, which produce mitochondrial ROS (mROS) to destruct the engulfed bacteria in macrophages[4,5]. Indeed, we observed that much fewer mitochondria were localized together with bacteria in Piezo1-deficient BMDMs than in wild-type cells during GFP–*E. coli* infection, while Yoda1 treatment enhanced the mitochondrion–phagosome juxtaposition in wild-type BMDMs, but not in Piezo1-deficient BMDMs, suggesting that Piezo1 did affect the mitochondrion–phagosome juxtaposition (Fig. 3e, f). We then measured the ROS levels in wild-type or Piezo1-deficient BMDMs using ROS-sensitive dyes (mitoSOX to measure the mROS superoxide, and CellROX to measure total cellular $H_2O_2$) and flow cytometry. The results showed that loss of Piezo1 diminished

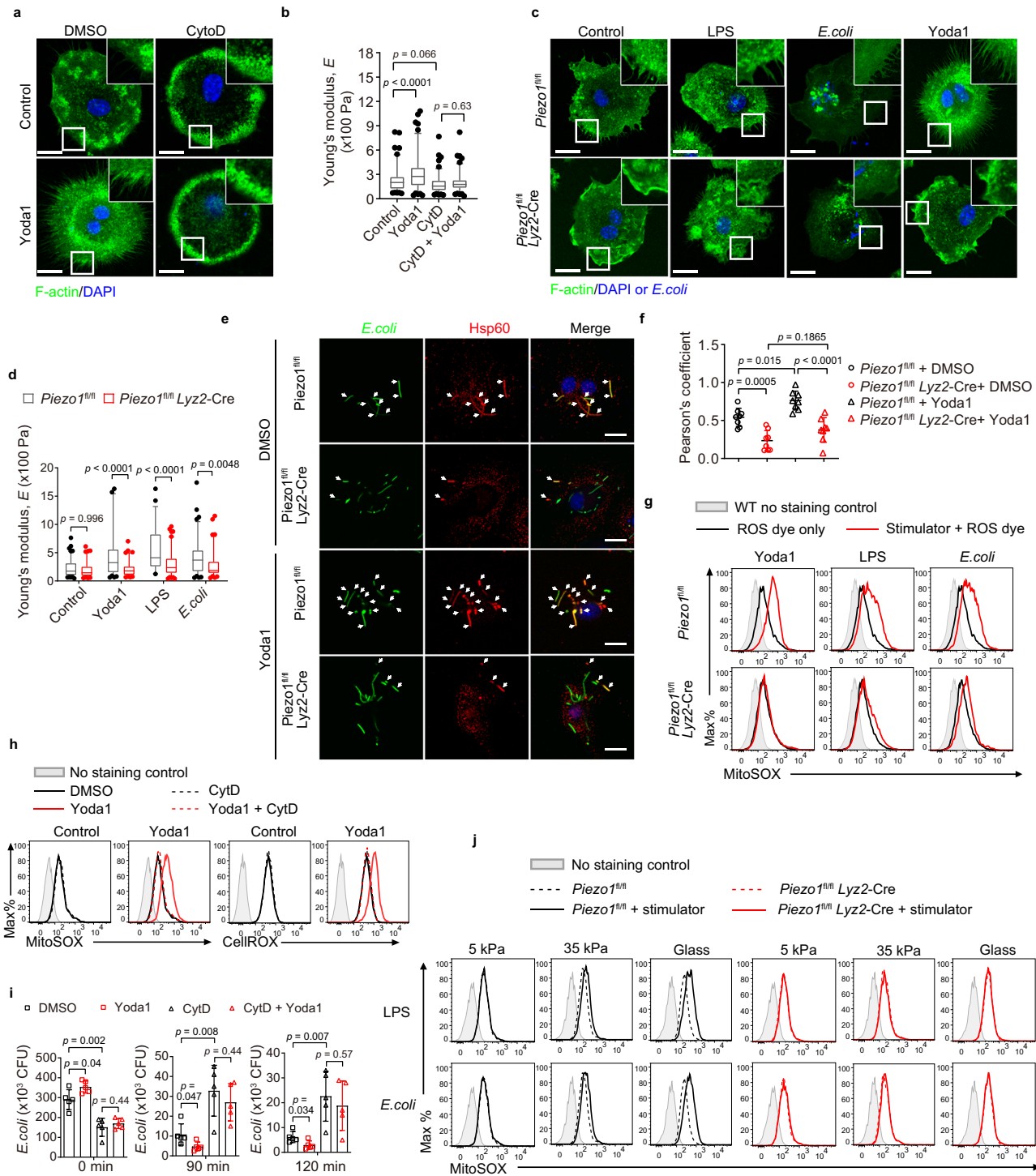

the production of mROS and cellular ROS in BMDMs treated with LPS, or infected with *E. coli* (Fig. 3g, Supplementary Fig. 3a). Notably, Yoda1-treated wild-type BMDMs, but not Piezo1-deficient BMDMs exhibit substantial induction of mROS and cellular ROS (Fig. 3g). Furthermore, CytD treatment that disrupts actin polymerization abrogated ROS production (Fig. 3h), and diminished the Yoda1-induced phagocytosis and bactericidal activity in BMDMs (Fig. 3h, i). Similarly, upon LPS stimulation or bacterial infection, the induction of mROS and cellular ROS was completely absent in BMDMs cultured on 5 kPa soft matrix

regardless of Piezo1 presenting (Fig. 3j, Supplementary Fig. 3b). In contrast, the mROS or cellular ROS could be induced in *Piezo1*fl/fl BMDMs cultured on 35 kPa matrix with a tiny amount production, or on glass matrix (much stiffer matrix) with much more amount production upon various stimulations, but ROS induction were diminished in *Piezo1*fl/fl *Lyz2*-Cre BMDMs cultured on any types of matrix (Fig. 3j, Supplementary Fig. 3b). These results suggested that Piezo1 is critical for the mitochondrion–phagosome juxtaposition and the bactericidal ROS production in macrophages during the infection.

**Fig. 3 Piezo1 is critical for cytoskeleton remodeling and ROS production. a** Confocal microscopy of BMDMs were pretreated with DMSO or cytochalasin D (CytD, 2 μM) for 1 h, followed by Yoda1 (5 μM) stimulation for 30 min, then immunostained with anti-F-actin (green) and counterstained with DAPI (blue). Outlined areas are enlarged in top right corners. Scale bars, 20 μm. **b** The Young's modulus of BMDMs treated with DMSO vehicle ($n = 105$), Yoda1 ($n = 110$), cytoD ($n = 110$), or Yoda1 plus CytD ($n = 108$) as indicated in (**a**). **c** Confocal microscopy of *Piezo1*[fl/fl] or *Piezo1*[fl/fl] *Lyz2*-Cre BMDMs treated with LPS (1 μg ml$^{-1}$), *E. coli* (MOI, 20), Yoda1 (5 μM) for 30 min, then immunostained with anti-F-actin (green) and counterstained with DAPI (blue); outlined areas are enlarged in top right corners. Scale bars, 20 μm. **d** The Young's modulus was determined by AFM for *Piezo1*[fl/fl] or *Piezo1*[fl/fl] *Lyz2*-Cre BMDMs treated with PBS control ($n = 104, 98$), Yoda1 (5 μM, $n = 82, 81$), LPS (1 μg ml$^{-1}$, $n = 55, 105$), or *E. coli* ($n = 95, 93$). **e, f** Confocal microscopy of the distribution of mitochondrial networks in *Piezo1*[fl/fl] or *Piezo1*[fl/fl] *Lyz2*-Cre BMDMs after infection with GFP–*E. coli* (green); nuclei were counterstained with DAPI (blue); arrows indicate colocalization of *E. coli* (green) and Hsp60 (red). Scale bars, 20 μm (**e**). Pearson's correlation coefficient values for colocalization of *E. coli* and Hsp60 in BMDMs. The average Pearson's correlation coefficients were calculated from eight randomly selected infected cells in each group ($n = 8$ cells examined) (**f**). **g** Flow cytometry analyzing mROS production by *Piezo1*[fl/fl] or *Piezo1*[fl/fl] *Lyz2*-Cre BMDMs stimulated with Yoda1 (5 μM) for 30 min, LPS (1 μg ml$^{-1}$) for 3 h, or *E. coli* (MOI, 20) for 1 h, followed by staining with MitoSOX. **h** Flow cytometry analyzing mROS and cellular ROS production by BMDMs treated with CytD or Yoda1 as indicated in (**a**). **i** Pathogen burden in BMDMs pretreated with DMSO or CytD (2 μM, 1 h), followed by Yoda1 (5 μM, 30 min) and then *E. coli* infection (MOI, 20) for 0, 90, or 120 min ($n = 5$ independent samples). **j** Flow cytometry analyzing mROS production by *Piezo1*[fl/fl] or *Piezo1*[fl/fl] *Lyz2*-Cre BMDMs treated LPS (1 μg ml$^{-1}$) for 3 h or *E. coli* (MOI, 20) for 1 h on different stiffness culture matrix of 5 kPa, 35 kPa, or glass, followed by staining with MitoSOX. In **b, d**, data are represented as boxplots where the middle line is the median, the lower and upper boundaries correspond to the first and third quartiles, whiskers above and below the box indicate the 5th and 95th percentiles, and points above and below the whiskers indicate outliers outside the 5th and 95th percentiles. Data are presented as mean +/− SD (**f, i**). The *p*-values of two-way ANOVA in (**b**), (**d**), (**f**) and two-tailed unpaired Student's *t* test in (**i**) are indicated. Data are representative of three independent experiments with similar results. Source data are provided as a Source data file.

**Piezo1 modulates cytoskeleton rearrangement via Rac1**. Previous studies have shown that cellular actin cytoskeleton rearrangements and filopodia formation are modulated by the small GTPase family proteins that switch between an active GTP-bound form and an inactive GDP-bound state to regulate their activity[36]. We then determined the activation status of the representative molecules of the small GTPases families, i.e., Rho/Rac/Cdc42 family (Rac1, Cdc42, and RhoA), Ras family (Ras and Rap2), and Sar1/Arf family (Arf6), in BMDMs upon the stimulation of Yoda1. Interestingly, we found that only Rac1 was activated by Yoda1 as estimated by pull-down with GST-tagged PAK70-106 in BMDMs treated with Yoda1 for 10 or 30 min (Fig. 4a and Supplementary Fig. 4a-d). Although the abundance of total Rac1 in Piezo1-deficient BMDMs was similar to that in wild-type BMDMs, the amount of active form of Rac1 (i.e., Rac-GTP) was much lower in Piezo1-deficient BMDMs than in wild-type BMDMs (Fig. 4b). Yoda1 treatment further increased the amount of Rac1-GTP in wild-type but not in Piezo1-deficient BMDMs (Fig. 4b). Consistently, Yoda1 triggered ROS production (Fig. 4c) and enhancement of bacterial phagocytosis and killing (Fig. 4d) were also diminished in Rac1-deficient BMDMs. These results suggested that Piezo1 augments bacterial phagocytosis and ROS production through Rac1-mediated signaling in macrophage.

To determine whether the defective activation of Rac1 is responsible for the impaired immune defense in Piezo1-deficient BMDMs, we crossed *Piezo1*[fl/fl] *Lyz2*-Cre mice with C57BL/6-Gt (ROSA)26Sor[tm9(Rac1*,EGFP)Rsky/J] mice, in which the expression of a constitutively active form of Rac1 (Rac1[G12V]) is induced in the presence of Cre recombinase, to generate myeloid cell-specific *Piezo1* knockout and *Rac1*[G12V] transgenic mice. BMDMs from *Piezo1*[fl/fl] *Rac1*[G12V] *Lyz2*-Cre mice displayed normal organization of F-actin and filopodia formation or actin accumulation at sites of bacteria upon LPS stimulations or *E. coli* infection (Fig. 4e). Consistently, AFM measurement showed that overexpression of Rac[G12V] in Piezo1-deficient BMDM resulted in higher cellular stiffness compared with that of Piezo1-deficient BMDMs with or without *E. coli* infection (Fig. 4f). In addition, overexpression of Rac[G12V] restored the colocalization of bacteria with the mitochondria during the GFP–*E. coli* infection (Fig. 4g, h), the production of mROS and cellular ROS in response to LPS treatment or *E. coli* infection (Fig. 4i and Supplemental Fig. 4e), as well as the ability of bacterial phagocytosis and clearance (Fig. 4j)

in Piezo1-deficient BMDMs. These results demonstrated that the activation of Rac1 is a critical downstream effect of Piezo1 for macrophages in the defense against infection.

**Mst kinases are crucial for Piezo1-mediated Rac1 activation**. Previously, we reported that Hippo kinases Mst1 and Mst2 (Mst1/2) positively regulate phagocytic phagosome–mitochondrion juxtaposition and the induction of phagosomal mROS through the small GTPase Rac1-mediated cytoskeletal reorganization[5]. Interestingly, loss of Mst1/2 or Mst1/2 kinases inhibitor treatment abrogated filopodia formation or actin accumulation at sites of bacteria induced by LPS or the Piezo1 agonist Yoda1 treatment or *E. coli* infection (Fig. 5a, b). Consistently, enhancement of cell stiffness measured by AFM in Yoda1-treated BMDMs was abrogated by the depletion of Mst1/2 (Fig. 5c) and the increased amount of Rac1-GTP in wild-type BMDMs upon Yoda1 treatment was not observed in Mst1/2-deficient BMDMs (Fig. 5d). These results suggested that kinases Mst1/2 might be involved in the regulation of Piezo1-mediated Rac1 activation and filopodia formation. Indeed, Yoda1 treatment robustly increased the phosphorylation levels of Mob1, a physiological substrate of the kinases Mst1/2 in wild-type BMDMs, but not in Piezo1-deficient BMDMs (Fig. 5e). In addition, the enhanced Mob1 phosphorylation in BMDMs upon the treatment of LPS or *E. coli*, was also diminished in Piezo1-deficient BMDMs, whereas the LPS or *E. coli* induced activation of mitogen-activated protein kinases, such as p38 and Jnk, was not affected by deleting Piezo1 (Fig. 5f, g). Consistently, ample amount of mROS or cellular ROS induced by Yoda1-triggered Piezo1 activation (Fig. 5h) and Yoda1-promoted phagocytosis and bacterial clearance (Fig. 5i) were also diminished in Mst1/2-deficient macrophages. Taken together, these data demonstrated that Piezo1 regulates macrophage cytoskeleton reorganization and bactericidal activity through the Mst1/2-Rac1 axis.

**Piezo1 activates kinases Mst1 and Mst2 through CaMKII**. As an ion channel, Piezo1 facilitates calcium influx into the cytoplasm[37,38]. Activation of Piezo1 by Yoda1 results in rapid calcium influx in wild-type BMDMs, but not in Piezo1-deficient BMDMs (Fig. 6a). Interestingly, the LPS-induced calcium influx was also dramatically suppressed in Piezo1-deficient BMDMs, indicating that the ion channel Piezo1 is important for the induction of calcium influx by TLR4 signaling (Fig. 6a). Previous

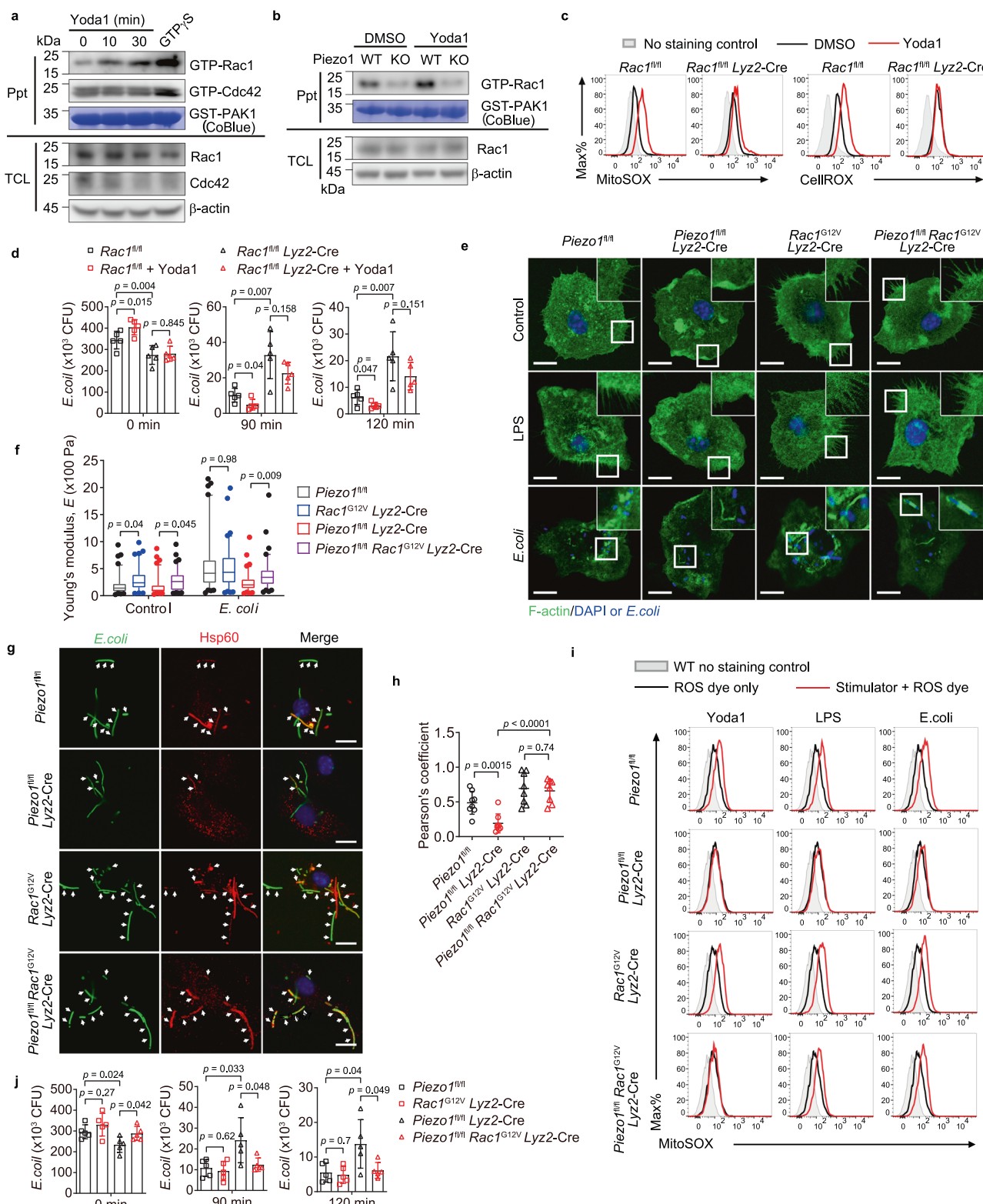

F-actin/DAPI or *E.coli*

study also highlighted the importance of calcium signaling in macrophage activation[39]. We then asked whether Piezo1-dependent calcium influx induces the activation of Mst1/2-Rac1 axis during the infection. First, we determined the effects of the Piezo1 agonist Yoda1 on Mst1/2 activation in BMDMs cultured with calcium-free media. The result showed that removal of extracellular calcium effectively inhibited Yoda1-induced Mst1/2 activation, shown as diminished Mob1 phosphorylation (Fig. 6b).

Consistently, elimination of intracellular calcium by calcium chelator BAPTA-AM treatment also effectively disrupted the Yoda1-induced Mst1/2 activation in a time-dependent manner and a dose-dependent manner (Fig. 6c–e), and this effect can only be observed in wild-type BMDMs, but not in Piezo1-deficient BMDMs (Fig. 6c). These results suggested that calcium was an important second messenger of Piezo1 signaling to activate kinases Mst1/2.

**Fig. 4 Piezo1 modulates cytoskeleton rearrangement via Rac1. a, b** Immunoblot analysis of the association of active (GTP-bound) Rac1 and Cdc42 with GST-PAK[70-106] in lysates of wild-type BMDMs treated with Yoda1 (5 μM) for 0, 10, or 30 min (**a**) or the association of active (GTP-bound) Rac1 in lysates of *Piezo1*[fl/fl] or *Piezo1*[fl/fl] *Lyz2*-Cre BMDMs treated with Yoda1 (5 μM) for 30 min (**b**); then incubated with GST-PAK[70-106] (GST-PAK); CoBlue, staining of GST-PAK[70-106] with Coomassie blue. **c** Flow cytometry analyzing mROS and cellular ROS production of BMDMs from *Rac1*[fl/fl] or *Rac1*[fl/fl] *Lyz2*-Cre mice treated with DMSO or Yoda1 (5 μM) for 30 min, followed by staining with MitoSOX or CellROX. **d** Pathogen burden in *Rac1*[fl/fl] or *Rac1*[fl/fl] *Lyz2*-Cre BMDMs treated with or without Yoda1 (5 μM, 30 min), followed by *E. coli* infection (MOI, 20) for 0, 90, or 120 min (*n* = 5 independent samples). **e** Confocal microscopy of *Piezo1*[fl/fl] or *Piezo1*[fl/fl] *Lyz2*-Cre, *Rac1*[G12V] *Lyz2*-Cre or *Piezo1*[fl/fl] *Rac1*[G12V] *Lyz2*-Cre BMDMs treated with control PBS, LPS (1 μg ml[−1]), or *E. coli* (MOI, 20) for 30 min, then immunostained with anti-F-actin (green) and counterstained with DAPI (blue); outlined areas are enlarged in top right corners. Scale bars, 20 μm. **f** The Young's modulus of *Piezo1*[fl/fl], *Piezo1*[fl/fl] *Lyz2*-Cre, *Rac1*[G12V] *Lyz2*-Cre or *Piezo1*[fl/fl] *Rac1*[G12V] *Lyz2*-Cre BMDMs treated with control PBS (*n* = 95, 94, 97, or 93) or *E. coli* (*n* = 90, 83, 93, or 88) for 30 min; data are represented as boxplots where the middle line is the median, the lower and upper boundaries correspond to the first and third quartiles, whiskers above and below the box indicate the 5th and 95th percentiles, and points above and below the whiskers indicate outliers outside the 5th and 95th percentiles. The *p*-values of two-way ANOVA are indicated. **g, h** Confocal microscopy of the distribution of mitochondrial networks in *Piezo1*[fl/fl], *Piezo1*[fl/fl] *Lyz2*-Cre, *Rac1*[G12V] *Lyz2*-Cre, or *Piezo1*[fl/fl] *Rac1*[G12V] *Lyz2*-Cre BMDMs after GFP–*E. coli* (green) infection; nuclei were counterstained with DAPI (blue) (**g**); arrows indicate colocalization of *E. coli* (green) and Hsp60 (red). Scale bars, 20 μm. Pearson's correlation coefficient values for colocalization of *E. coli* and Hsp60 in BMDMs. The average Pearson's correlation coefficients were calculated from eight randomly selected infected cells in each group (*n* = 8 cells examined) (**h**). **i** Flow cytometry analyzing mROS production by *Piezo1*[fl/fl], *Piezo1*[fl/fl] *Lyz2*-Cre, *Rac1*[G12V] *Lyz2*-Cre or *Piezo1*[fl/fl] *Rac1*[G12V] *Lyz2*-Cre BMDMs stimulated with Yoda1 (5 μM) for 30 min, LPS (1 μg ml[−1]) for 3 h, or *E. coli* (MOI, 20) for 1 h, followed by staining with MitoSOX. **j** Pathogen burden (CFU) in *Piezo1*[fl/fl], *Piezo1*[fl/fl] *Lyz2*-Cre, *Rac1*[G12V] *Lyz2*-Cre, or *Piezo1*[fl/fl] *Rac1*[G12V] *Lyz2*-Cre BMDMs infected with GFP–*E. coli* for 0, 90, or 120 min (*n* = 5 independent samples). Data are presented as mean +/− SD and the *p*-values of two-tailed unpaired Student's *t* test in (**d**), (**h**), (**j**) are indicated. Data are representative of three independent experiments with similar results. Source data are provided as a Source data file.

We then wondered whether calcium-dependent protein kinases such as PKC, FAK, and Ca$^{2+}$/calmodulin (CaM)-dependent kinases (CaMKs) might be involved in regulating the activity of kinases Mst1/2. To this end, BMDMs were pretreated with various inhibitors to block the corresponding kinases, respectively, and we found that only KN-62 (a pan CaMKs inhibitor) diminished the activation of Mst1/2 by Yoda1 (Fig. 6f). Co-immunoprecipitation assays revealed that Mst1 could interact with all the isoforms of CaMKI, II, IV, among which CaMKII has the most potent association (Fig. 6g). In addition, the phosphorylation of Mst1K/R, which cannot be auto-phosphorylated, was observed in the presence of CaMKII, suggesting Mst1 is a substrate of CaMKII (Fig. 6h). We then chose autocamtide-2-related inhibitory peptide (TFA), a highly specific and potent inhibitor of CaMKII, to perform the following function assays. Consistently, inhibition of CaMKII abrogates Yoda1-induced Rac1-GTP charging (Fig. 6i), cell stiffness (Fig. 6j), ROS induction (Fig. 6k), mitochondrion–phagosome juxtaposition (Fig. 6l, m), and bacterial clearance (Fig. 6n). Taken together, these results established that the mechanically activated ion channel Piezo1 regulates macrophages bactericidal activity through calcium signal-induced activation of the CaMKII-Mst1/2-Rac1 axis (Supplementary Fig. 5).

## Discussion

The stiffer microenvironments of acute and chronic inflammation that macrophages are often present in highlights the necessity to consider mechanical sensing signals that may influence macrophage functions. In this manuscript, we revealed that the mechanically activated ion channel Piezo1 might function as a co-receptor of TLR4 to promote cytoskeleton remodeling, intracellular organelle trafficking, and innate response. We found that LPS stimulation or *E. coli* infection promotes the engagement of the corresponding TLR4 receptor with the mechanical sensor Piezo1 to induce calcium influx. Moreover, Piezo1 is required to enhance phagocytosis and mitochondrion–phagosome juxtaposition for ROS generation. Piezo1 deficiency in macrophage impairs calcium influx, F-actin reorganization, ROS production, and bactericidal activity. We further revealed that Piezo1 regulates macrophage bactericidal activity through calcium signal-induced activation of the CaMKII-Mst1/2-Rac axis. Inhibition of CaMKII or knockout of either Mst1/2 or Rac1 results in reduced

bacterial clearance, phenocopying Piezo1 deficiency. These results provide insight into the mechanism and function of Piezo1 governing mechanical transduction system during the infection.

Piezo1 has a remarkably dynamic pattern of expression and localization, and changes in levels and/or subcellular redistribution modulate Piezo1 activity. Recent study by Solis et al. showed that Piezo1 recognition of the cyclical hydrostatic pressure microenvironment in the lung modulates inflammatory response[31]. Our study showed that Piezo1 coordinates TLR4 to translate extracellular mechanical and chemical cues into cellular signal transduction inside the innate cell. Our findings demonstrated a critical role of Piezo1 in remodeling cytoskeleton for phagocytosis, ROS production, and bactericidal activity in response to various infectious cues. In sum, these two complimentary studies shed light on the essential role of Piezo1 in macrophage mechanophysiology and innate immunity. Piezo1 has also been shown to regulate other immune cell activation[33,35]. Likely, Piezo1 might also respond to other stimuli, such as chemokines and cytokines, and coordinate their corresponding receptors in immune cells. Thus, our findings provide critical insight into how soluble stimuli take advantage of cellular mechanical transduction system to achieve immune responses.

Hippo kinase Mst1 deficiency in humans and mice results in a complex combined immunodeficiency syndrome[6,7,12,40–42]. We and others previously showed that kinases Mst1/2 regulate the migration, adhesion, and activation of T cells, and are also essential in innate immunity[9,10,43,44]. Recent studies have reported that epithelium cell stretching leads to activation of the downstream transcription cofactors YAP and TAZ of Hippo signaling, which in turn promote cell proliferation[15,45–53]. However, the expression levels of YAP and TAZ are quite low or non-detectable in innate immune cells indicating that they may be unique mechanic effectors for epithelium cells. In this manuscript, we showed that Piezo1 remodels macrophage rigidity through Mst1/2-Rac1 axis. Several studies have highlighted the significance of Rac in modulating cell morphology[36,54,55]. Among the three highly conserved isoforms of Rac, Rac1 is ubiquitous in all mammalian cells including macrophages, whereas Rac2 is only found in hematopoietic and endothelial cells. Deletion of Rac1 in mice has been shown to alter macrophage cell shape. Comparing to wild-type BMDMs, Rac1-deficient BMDMs display elongated morphologies and reduced spread area. Moreover, the

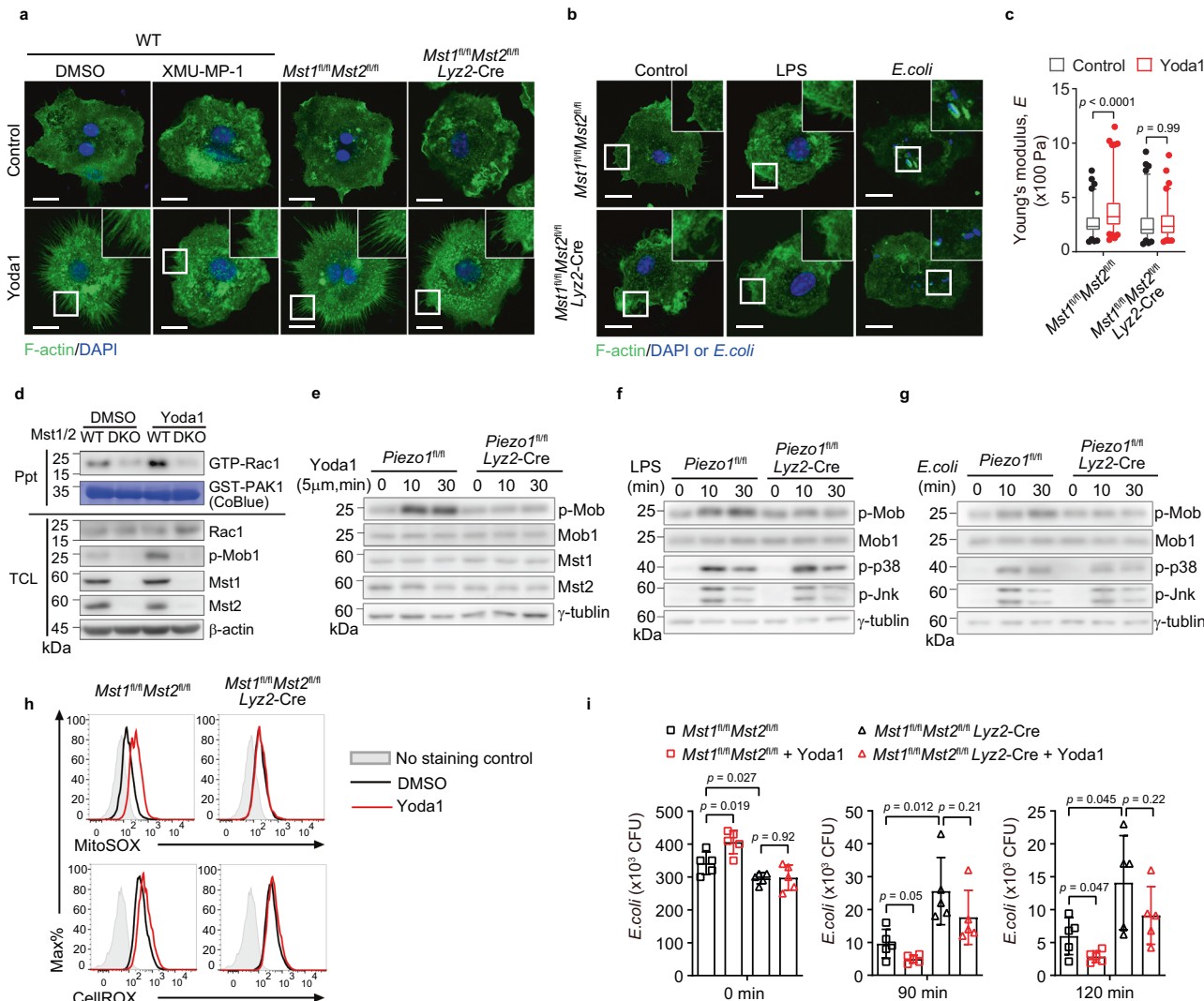

**Fig. 5 Mst1 and Mst2 are required for Piezo1-mediated Rac1 activation. a** Confocal microscopy of *Mst1*fl/fl*Mst2*fl/fl or *Mst1*fl/fl*Mst2*fl/fl *Lyz2*-Cre BMDMs treated with the vehicle DMSO, Yoda1 (5 μM), or XMU-MP-1 (3 μM) for 30 min, then immunostained with anti-F-actin (green) and counterstained with DAPI (blue); outlined areas are enlarged in top right corners. Scale bars, 20 μm. **b** Confocal microscopy of *Mst1*fl/fl *Mst2*fl/fl or *Mst1*fl/fl *Mst2*fl/fl *Lyz*-Cre BMDMs treated with the vehicle PBS, LPS (1 μg ml⁻¹), or *E. coli* (MOI, 20), then immunostained with anti-F-actin (green) and counterstained with or without DAPI (blue); outlined areas are enlarged in top right corners. Scale bars, 20 μm. **c** The Young's modulus of *Mst1*fl/fl*Mst2*fl/fl and *Mst1*fl/fl*Mst2*fl/fl *Lyz2*-Cre BMDMs treated with DMSO vehicle ($n = 97$, $n = 91$) or Yoda1 (5 μM) ($n = 104$, $n = 90$) for 30 min. Data are represented as boxplots where the middle line is the median, the lower and upper boundaries correspond to the first and third quartiles, whiskers above and below the box indicate the 5th and 95th percentiles, and points above and below the whiskers indicate outliers outside the 5th and 95th percentiles. The $p$-values of two-way ANOVA are indicated by bracketing. **d** Immunoblot analysis of the association of active (GTP-bound) Rac1 with GST-PAK⁷⁰⁻¹⁰⁶ in lysates of *Mst1*fl/fl*Mst2*fl/fl or *Mst1*fl/fl *Mst2*fl/fl *Lyz2*-Cre BMDMs treated with Yoda1 (5 μM) for 30 min, then incubated with GST-PAK⁷⁰⁻¹⁰⁶ (GST-PAK); CoBlue, staining of GST-PAK⁷⁰⁻¹⁰⁶ with Coomassie blue. **e** Immunoblot analysis of phosphorylated (p)-Mob1, Mob1, Mst1, Mst2, and γ-tubulin in *Piezo1*fl/fl or *Piezo1*fl/fl *Lyz2*-Cre BMDMs stimulated for 0, 10, or 30 min (above lanes) with Yoda1 (5 μM). **f, g** Immunoblot analysis of phosphorylated p-p38, p-Jnk and p-Mob1, Mob1, and γ-tubulin in *Piezo1*fl/fl or *Piezo1*fl/fl *Lyz2*-Cre BMDMs stimulated with LPS (1 μg ml⁻¹) (**f**) or *E. coli* (MOI, 20) (**g**) for 0, 10, or 30 min. **h** Flow cytometry analyzing mROS and cellular ROS production of *Mst1*fl/fl*Mst2*fl/fl or *Mst1*fl/fl*Mst2*fl/fl *Lyz2*-Cre BMDMs treated with DMSO or Yoda1 (5 μM) for 30 min, followed by staining with MitoSOX or CellROX. **i** Pathogen burden (CFU) in *Mst1*fl/fl*Mst2*fl/fl or *Mst1*fl/fl*Mst2*fl/fl *Lyz2*-Cre BMDMs treated with or without Yoda1 (5 μM, 30 min), followed by GFP–*E. coli* infection for 0, 90, or 120 min ($n = 5$ independent samples). Data are presented as mean +/− SD and the $p$-values of two-tailed unpaired Student's $t$ test are indicated. Data are representative of three independent experiments with similar results. Source data are provided as a Source data file.

importance of Rac-GTP in phagocyte function is illustrated by a human immunodeficiency syndrome characterized by severe bacterial infections that arise from a mutation in the gene encoding Rac2 that generates the D57N substitution of Rac2 (Rac2^D57N), which results in constitutive binding of GDP, accompanied by impaired ROS production, in phagocytes[56,57].

Thus, Rac could be a critical downstream effector of Piezo1 for cytoskeleton reorganization and ROS generation.

Macrophages from different tissues within the same host may have different sensitivity to mechanical cues[32]. For example, alveolar macrophages and osteoclasts naturally undergo more mechanical loading in the body, and therefore may be more sensitive to physical

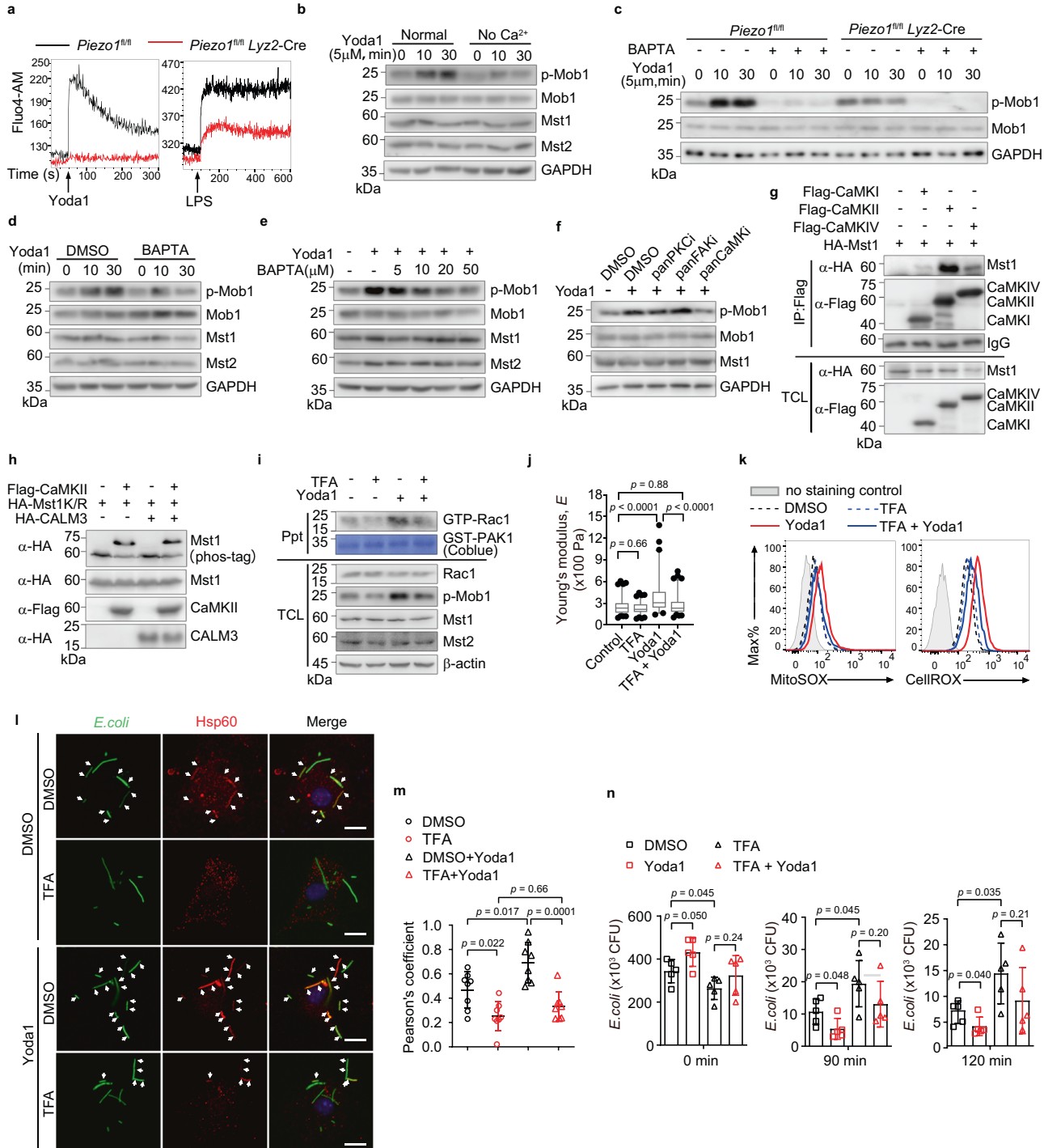

stimuli. In addition, ECMs from different tissues can have dramatically different stiffness and architecture, which may result in differential responsiveness to physical signals by macrophages in those tissues. There is likely a range of ECM stiffness and architecture that best supports tissue homeostasis and physiological macrophage function. In addition, macrophages are notorious for being very heterogeneous. It might be important to increase or decrease the mechanical property of injured tissues according to the needs for curing different diseases. Thus, a lot of work is still needed in elucidating the underlying mechanism and biological functions for Piezo1-mediated mechanobiology of macrophages from various sources. In the future, it will be important to determine the role of mechanical sensor Piezo1 in immune cell polarization and tumor

immunity in cancer microenvironment. Discovery of efficient Piezo1 agonist and antagonist in vivo will be important for immune-relevant diseases treatment.

## Methods

**Animals**. The conditional knockout of *Mst1* and *Mst2* has been described previously[58]. Wild-type C57BL/6 mice, *Piezo1*P1-tdT mice (029214), *Piezo1*flox mice (029213), *Piezo2*-EGFP-IRES-Cre mice (027719), *Rac1*flox mice (005550), Rosa26-LSL-*Rac1*G12V (012361) mice, C57BL/10ScNJ (*Tlr4*lps-del, 003752), and *Lyz2*-Cre mice (004781) were originally from the Jackson Laboratory. All Mice were housed under specific pathogen-free conditions with a 12 h light/dark cycle, at a temperature of 22 +/− 2 °C and a relative humidity of 50 +/− 5%, and were fed with a standard mouse chow diet at the Xiamen University Laboratory Animal Center. These mouse experiments were approved by the Institutional Animal Care and Use

**Fig. 6 Piezo1 activates kinases Mst1/2 through CaMKII. a** Calcium influx over time in *Piezo1*[fl/fl] or *Piezo1*[fl/fl] *Lyz2*-Cre BMDMs treated with Yoda1 (5 μM) or LPS (1 μg ml$^{-1}$). **b** Immunoblot analysis of phosphorylated (p)-Mob1, Mob1, Mst1, Mst2, and GAPDH in BMDMs pretreated with DMEM with (normal) or without calcium (no Ca$^{2+}$) for 2 h and then treated with Yoda1 (5 μM) for 0, 10, 30 min. **c** Immunoblot analysis of p-Mob1, Mob1, and GAPDH in *Piezo1*[fl/fl] or *Piezo1*[fl/fl] *Lyz2*-Cre BMDMs cells pretreated with BAPTA-AM (20 μM) for 2 h, then followed by Yoda1 stimulation for 0, 10, or 30 min. **d, e** Immunoblot analysis of phosphorylated p-Mob1, Mob1, Mst1, Mst2, and GAPDH in BMDMs pretreated with BAPTA-AM (20 μM) for 2 h and then treated with Yoda1 (5 μM) for 0, 10, 30 min (**d**), or pretreated with BAPTA-AM (0, 5, 10, 20, 50 μM) for 2 h and then treated with Yoda1 (5 μM) for 30 min (**e**). **f** Immunoblot analysis of p-Mob1, Mob1, Mst1, and GAPDH in BMDMs pretreated for 2 h with PKC inhibitor (Bisindolylmaleimide I, 4 μM), FAK inhibitor (TAE226, 10 μM), CaMKs inhibitor (KN-62, 10 μM), then followed by Yoda1 (5 μM) stimulation for 30 min. **g** Immunoblot analysis of 293T cells expressing various combinations of HA-tagged Mst1 and Flag-tagged CaMKI, CaMKII, or CaMKIV, immunoprecipitated with anti-Flag and analyzed by immunoblot with anti-HA (α-HA) or anti-Flag (α-Flag); below, immunoblot analysis of total cell lysates (TCL) without immunoprecipitation. **h** Phos-tag and SDS-PAGE analysis of HA-tagged kinase-inactive Mst1 (Mst1K/R) expressed together with Flag-tagged CaMKII and/or HA-tagged Calmodulin-3 (HA-CALM3) in 293T cells by immunoblotting with anti-HA (α-HA) or anti-Flag (α-Flag) antibodies. **i** BMDMs Immunoblot analysis of the association of active (GTP-bound) Rac1 with GST-PAK$^{70-106}$ in lysates of BMDMs pretreated with TFA (10 μM) for 2 h and then treated with Yoda1 (5 μM) for 30 min. **j** The Young's modulus of BMDMs treated with DMSO vehicle (n = 97), TFA (10 μM) (n = 99), Yoda1 (5 μM) (n = 90), or TFA plus Yoda1 (n = 90) as in (**h**). Data are represented as boxplots where the middle line is the median, the lower and upper boundaries correspond to the first and third quartiles, whiskers above and below the box indicate the 5th and 95th percentiles, and points above and below the whiskers indicate outliers outside the 5th and 95th percentiles. **k** mROS and cellular ROS production by BMDMs with the indicated treatment, followed by staining with MitoSOX or CellROX. **l, m** BMDMs pretreated with DMSO or TFA (10 μM) for 2 h and then infection with GFP–*E. coli* (MOI, 20) (green). Confocal microscopy of the distribution of mitochondrial networks after 30 min of bacterial infection, nuclei were counterstained with DAPI (blue); arrows indicate colocalization of *E. coli* (green) and Hsp60 (red) (**l**), Pearson's correlation coefficient values for colocalization of *E. coli* and Hsp60 in BMDMs. The average Pearson's correlation coefficients were calculated from eight randomly selected infected cells in each group (n = 8 cells examined) (**m**). Scale bars, 20 μm. **n** Pathogen burden (CFU) in BMDMs treated as indicated, followed by infection with GFP–*E. coli* for 0, 90, or 120 min (n = 5 independent samples). Data are presented as mean +/− SD (**m**, **n**). The *p*-values of two-way ANOVA in (**j**), (**m**) and two-tailed unpaired Student's *t* test in (**n**) are indicated by bracketing. Data are representative of three independent experiments with similar results. Source data are provided as a Source data file.

---

Committee and were in strict accordance with good animal practice as defined by the Xiamen University Laboratory Animal Center.

**Chemicals and reagents**. LPS (L4391), GTPγS (G8634), and Cytochalasin D (C8273) were from Sigma-Aldrich. Bisindolylmaleimide I (S7208), TAE226 (S2820), and KN-62 (S7422) were from Selleck. TFA (P0214A) was from Med-ChemExpress (MCE). BAPTA-AM (2787) and Yoda1 (5586) were from TOCRIS. The Phos-tag TM Acrylamide AAL-107 was from the NARD Institute.

**Cell culture**. The 293T cell lines (from the American Type Culture Collection) were tested for mycoplasma contamination and were found to be negative, then were cultured in DMEM supplemented with 10% FBS and 1× penicillin-streptomycin (Invitrogen). For BMDMs, the femur and tibia were collected from mice of each genotype, and bone marrow cells were flushed with complete DMEM containing 50 mg ml$^{-1}$ streptomycin and 10% FBS. Erythrocytes were removed via treatment with red blood cell lysis buffer, and the cell suspensions were filtered through a 40-μm cell strainer for the removal of any cell clumps. The single-cell suspensions were then cultured for 1 h at 37 °C, and non-adherent cells were collected and re-plated in complete DMEM with 25% medium conditioned by recombinant mouse M-CSF (carrier-free, 50 ng ml$^{-1}$). For full differentiation of BMDMs, the cells were cultured for an additional 8 days with replacement of the medium every 2 days. All cells were CD11b$^+$F4/80$^+$ when analyzed by flow cytometry.

**Latex bead phagocytosis assay**. 641 nm 1.75-μm Fluoresbrite carboxy latex microparticles (17797-1, Polysciences) were untreated or coated overnight at 4 °C with 30 μg/ml LPS in PBS. The beads were washed ten times in large volumes of PBS containing 1% FBS for removal of unbound LPS. BMDMs were cultured on coverslips in six-well dishes and then were put on ice for 10 min. Microparticles were added at a concentration of approximately three to five beads per cell and were allowed to settle on the cells for an additional 10 min on ice. The plates were warmed to 37 °C by adding warm medium for the appropriate time to allow bead phagocytosis followed by cellular fixation with 3.7% paraformaldehyde for 20 min at room temperature. The fixed cells were permeabilized with 0.1% Triton X-100 and then stained with anti-RFP (1:100 dilution; ab62341; Abcam) and anti-TLR4 (1:100 dilution; 19811-1-AP; Proteintech) for 30 min, followed by incubating with the secondary antibodies (Alexa Fluor 488-conjugated anti-mouse IgG (A21202) and Alexa Fluor 555-conjugated anti-rabbit IgG (A31572) (all from Invitrogen)) for another 1 h at 4 °C. Subsequently, the cells were washed with PBS three times and mounted with pure glycerin. All images were collected with a Precision DeltaVision-OMX Super-Resolution Microscope (GE OMX V4).

**Colocalization analysis**. For colocalization analysis, images were taken with a ×63 or ×100 oil objective, and the analysis was done with Image-Pro Plus software. The autothreshold function was used to process and analyze the colocalized pixels.

Pearson's correlation coefficient (Rr) was used to measure the degree of colocalization, with 3–5 images/group.

**Fluorescence resonance energy transfer (FRET) assay**. 293T cells were transfected with plasmids of mTagBFP-Piezo1 and GFP-TLR4, CFP and YFP, or BFP–GFP fusion protein. FRET analysis was performed according to the acceptor photobleaching method using a confocal microscope ZEISS LSM-780 (Carl Zeiss). Photobleach was done by 488 nm laser irradiation, repeated 200 times. The excitation wavelength was 405 and 488 nm for detecting fluorescent signal of mTagBFP and GFP[59,60], respectively. These emissions were collected at 440–480 nm and 500–560 nm for BFP and GFP, respectively. FRET efficiency was calculated according to the equation (ZEN black), FRET efficiency (%) = (D Post − D Pre)/D Post × 100. D Post: BFP signal after photobleaching; D Pre: BFP signal before photobleaching.

**Flow cytometry assay**. Single cells isolated from the bone marrow, spleen, lymph nodes, or peritoneal fluid were stained with the indicated fluorescence-conjugated antibodies for 20 min, washed, and then were resuspended with flow cytometry staining buffer (1% BSA in PBS) containing DAPI (Invitrogen, Carlsbad, CA, USA). Stained cells were analyzed with a BD LSRFortessa$^{TM}$ flow cytometer (BD Biosciences, Bedford, USA). Flow cytometry data were plotted and quantified using median fluorescence intensity (MFI) using FlowJo software (BD Biosciences) (Supplementary Fig. 6a and b). The fluorescence-conjugated antibodies, anti-CD11b (M1/70), anti-F4/80 (BM8), anti-Gr-1 (RB6-8C5), anti-CD11c (N418), anti-Ly-6A/E (E13-161.7), anti-CD117 (2B8), and anti-mouse Lineage Cocktail antibodies, were from BD Biosciences or Biolegend (San Diego, CA, USA).

**Measurement of ROS and mROS**. BMDMs were plated in non-tissue-culture-treated dishes and treated with Yoda1, TFA, LPS, or infected with bacteria as indicated. Samples were pretreated with CaMKs inhibitor (TFA, 10 μM) for 1 h or cytochalasin D (2 μM) for 1 h as needed, followed by the treatment of other stimulants or infected with bacteria as indicated. The concentrations of stimulants or bacteria were as follows: Yoda1 (5 μM), LPS (1 μg ml$^{-1}$), or *E. coli* (MOI, 20). The culture medium was removed and then the cells were washed with PBS and then incubated for 30 min at 37 °C with MitoSOX (for measurement of mROS superoxide; Invitrogen) or CellROX (for measurement of total cellular H$_2$O$_2$; Invitrogen) at a final concentration of 5 μM in serum-free DMEM (Invitrogen). The cells were washed with warmed PBS, removed from the plates by pipetting with cold PBS containing 1 mM EDTA, pelleted at 1500 r.p.m. for 3 min, immediately resuspended in cold PBS containing 1% FBS and analyzed by flow cytometry (Supplementary Fig. 6c).

**Probing intracellular calcium**. BMDMs (3 × 10$^6$) were stained with 4 μM Fluo-4 AM (Invitrogen), in DMEM for 30 min at 37 °C with gentle agitation. Following one wash in DMEM, the cells were resuspended at 1 × 10$^6$ cells ml$^{-1}$ and analyzed by flow cytometry (Supplementary Fig. 6c).

**Force indentation acquisition by AFM.** Mechanical properties of individual cells were measured using an atomic force microscope (Bio-resolve; Bruker, USA) mounted onto an inverted optical microscope (Nikon Eclipse Ti-E; Nikon, Japan) with a cell heater attachment. Force indentation measurements were carried out using in-house-prepared AFM colloidal probes (a spherical silica bead with the diameter of 9.4 μm glued on the cantilever by epoxy; the probe is AC40 from Bruker). Calibration measurements were performed before every experiment to determine the spring constant for each cantilever as described previously[61]. Cells were cultured overnight prior to force indentation measurements. During the experiment, cells were kept at 37 °C and in 1% HEPES buffered full medium to maintain pH levels. Indentations were performed at a loading force of 0.4 nN and a constant speed of 3 μm s$^{-1}$. The Young's modulus was derived by fitting a force–distance curve with the Hertzian spherical model[62]. Comparative studies between different populations were conducted using the same AFM probe and under the same conditions.

**CLP model.** Mice were anesthetized, and an abdominal incision was made for identification of the cecum. The distal one-third of the cecum was ligated with silk suture and was punctured once with a 22-gauge needle. A small amount of cecal content was extruded through the perforation. The peritoneum and skin were closed with a continuous suture after the cecum was returned to the abdomen. One milliliter saline was injected intraperitoneally for resuscitation. For sham-treated mice, all of the same steps were performed, except for ligation and puncture of the cecum.

**Measurement of bacterial loads in tissues.** Bacterial counts were performed on aseptically obtained liver, lung, kidney, and spleen. Twenty-four hours after CLP and sham surgery, mice were euthanized, and the skin of abdomen was cut open in the midline without injury to the muscle. Homogenates of liver, lung, kidney, and spleen samples were serially diluted in PBS and cultured on tryptose soy agar (TSA) blood agar plates. CFUs were counted after 18–24 h incubating 37 °C and the results were expressed as log10 of the number of CFU mL$^{-1}$. Values for CFU were expressed as colony counts per gram of tissue.

**Phagocytosis and bacteria-killing assay.** For FACS-based measurement of phagocytosis, a total of $1 \times 10^6$ BMDMs in PBS were cooled down to 4 °C for 30 min. Then, the cells were left uninfected or infected with heat-killed, FITC-labeled, and fresh mouse serum-opsonized E. coli (MOI, 20) for 20 min at 37 or 4 °C, after which they were washed extensively with cold PBS twice and fixed with 4% paraformaldehyde. The fluorescence of extracellular particles was quenched by replacing the medium with 0.2% Trypan blue in PBS, pH 5.5, shortly before the actual measurement with FACS.

For the in vitro bacterial killing assay, fresh overnight cultures of E. coli were suspended in PBS and opsonized with fresh mouse serum. The BMDMs were incubated with E. coli (MOI, 20) for the time points as described at 37 °C with intermittent shaking. After each time period, cells were lysed by adding distilled H$_2$O and diluted aliquots were spread on LB agar (E. coli) plates. The CFUs were counted after incubating the plates overnight at 37 °C.

**Hydrogel preparation for cell culture.** Polyacrylamide (PA) gels were made from 40% acrylamide and 2% bis-acrylamide mixed with 10% ammonium persulfate and 1% TEMED, where varying ratios of acrylamide and bis-acrylamide were used to create gels of known reproducible stiffness[63]. Gels were cast on glutaraldehyde-modified coverglasses. After polymerization, the gel was washed with PBS thoroughly to remove unreacted reagents. The PA gel surfaces were then conjugated with fibronectin using Sulfo-SANPAN under 365 nm ultraviolet exposure. Fibronectin was then added to the surface at 200 μgml$^{-1}$ to allow coupling to the gel through side-chain primary amines for 1 h at 37 °C.

**SDS-PAGE, Phos-tag SDS-PAGE, and immunoblot analysis.** Gels for SDS-PAGE or Phos-tag SDS-PAGE were prepared according to the manufacturer's instructions (NARD Institute). Separated proteins were transferred onto a PVDF membrane and then were identified by immunoblot analysis with the appropriate primary antibodies at a dilution of 1:1000 (or as otherwise stated below). Antibody to Mob1 phosphorylated at Thr35 (1:2000 dilution; 8699), anti-Mob1 (1:2000 dilution; 13730), anti-Mst1 (1:2000 dilution; 3682), anti-Mst2 (1:2000 dilution; 3952), anti-GAPDH (1:5000 dilution; 5174), antibody to phosphorylated p38 (1:3000 dilution; 9211), antibody to phosphorylated Jnk (1:3000 dilution; 4668), and anti-β-actin (1:5000 dilution; 8457) were from Cell Signaling Technology. Anti-Rac1 (66122-1-Ig), anti-HA Rabbit (51064-2-AP), and anti-Flag Rabbit (1:3000 dilution; 20543-1-AP) were from Proteintech. Anti-gamma Tubulin (1:5000 dilution; ab11316) was from Abcam and anti-Flag Mouse (1:5000 dilution; sc-166355) was from Santa Cruz Biotechnology. The protein bands were visualized with a SuperSignal West Pico Kit according to the manufacturer's instructions (Thermo Fisher Scientific Pierce).

**Confocal fluorescence microscopy.** For fluorescence analysis of F-actin, BMDMs seeded on glass coverslips in six-well dishes were incubated for 30 min with LPS, Yoda1, XMU-MP-1, or BFP–E. coli. The cells on coverslips were fixed for 10 min

with 3.7% paraformaldehyde and were washed three times for 5 min each with 0.1% Triton X-100 in PBS. F-actin in cells was stained with Actin-Tracker Green (Beyotime; C1033) in a solution containing 0.1% Triton X-100 and 5% BSA in PBS. After incubation for 1 h at room temperature, the cells were washed extensively, then were mounted with PBS and were mounted with Vectashield mounting medium containing DAPI or pure glycerin and imaged with a fluorescence microscope (Zeiss LSM-780).

For fluorescence analysis of other molecules, BMDMs seeded on glass coverslips in six-well dishes were incubated for 30 min with LPS, GFP–E. coli, or BFP–E. coli. The cells were fixed for 15 min at room temperature with 3.7% (vol/vol) paraformaldehyde, after which additional immunofluorescence staining was applied. For staining with anti-Hsp60 (1:100 dilution; 12165; Cell Signaling Technology), anti-RFP (1:100 dilution; ab62341; Abcam), anti-TLR4 (1:100 dilution; 19811-1-AP; Proteintech), fixed cells were rinsed with PBS and then were incubated for 10 min on ice with 0.2% Triton X-100 and 0.2% BSA in PBS. Following permeabilization, nonspecific binding in the cells was blocked by incubation for 30 min at room temperature with 0.02% Triton X-100 and 5% BSA in PBS, and cells were incubated for 1 h with specific primary antibodies (identified above). After three washes with PBS, the cells were incubated for another 1 h with secondary antibodies (Alexa Fluor 488-conjugated anti-mouse IgG (A21202), Alexa Fluor 555-conjugated anti-rabbit IgG (A31572) (all from Invitrogen). Subsequently, the cells were washed three times with PBS and were mounted with Vectashield mounting medium containing DAPI or pure glycerin. All images were collected with a confocal microscope (Zeiss LSM-780) or a Precision DeltaVision-OMX Super-Resolution Microscope (GE OMX V4).

**Quantitative real-time PCR.** For mRNA analysis, total mRNA was isolated from immortalized murine macrophage using RaPure Total RNA Kit (MAGEN, Cat. No. R4011-02), and cDNA was synthesized from 1 μg total RNA using 5X All-In-One RT Mastermix (Applied Biological Materials, Cat. No. G492). qPCR was performed using MonAmpChemoHS qPCR Mix (Monad, Cat. No. MQ00101) with the instrument CFX96 Touch Real-Time PCR Detection System (BioRAD). Primer pairs used in this study are listed in Supplementary Table 1.

**Assay of endogenous GTP-bound small GTPases.** BMDMs seeded in a 60 mm dish with 80% confluence were treated with Yoda1 for 0, 10, or 30 min, and then harvested using binding buffer (50 mM Tris-HCl, pH 7.5, 150 mM NaCl, 1% Triton X-100, 10 mg/ml leupeptin, 10 mM NaF, 2 mM Na$_3$VO$_4$, and 1 mM PMSF). Twenty micrograms of E. coli-purified GST-PAK1$^{70-106}$ (for Rac1 and Cdc42), GST-RTKN$^{RBD}$ (for RhoA), GST-Raf$^{RBD}$ (for Ras), GST-RhoGDS$^{RBD}$ (for Rap2), or GST-GGA3$^{RBD}$ (for Arf6) preloaded on GSH Sepharose was incubated with the precleared cell lysates at 4 °C for 2 h. After three washes with binding buffer, the amount of GTP-bound small GTPase to the beads was determined by indicated immunoblot after SDS-PAGE.

**Quantification and statistical analysis.** All data are representative of at least three independent experiments. All statistical analyses were performed using Prism6 (GraphPad). The data are presented as the mean +/− SD as indicated in the legends. The two-way ANOVA analysis was used for comparing cell stiffness or the colocalization of E. coli phagosomes with mitochondria among groups in one graph. The two-tailed unpaired Student's $t$ test was used for comparisons of bacteria number between groups in bacterial killing assays. Survival data were analyzed by the Kaplan–Meier statistical method. A $p$-value < 0.05 was considered statistically significant.

**Reporting summary.** Further information on research design is available in the Nature Research Reporting Summary linked to this article.

## Data availability
The data that support the findings of this study are available from the corresponding authors upon reasonable request. Source data are provided with this paper.

## Biological materials availability
All unique materials used are readily available from the authors or from the standard commercial sources.

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

## Acknowledgements

This work was supported by grants from National Key R&D Program of China (2017YFA0504502 to D.Z. and L.C.), The National Natural Science Foundation of China (81790254, 31625010, and 82021003 to D.Z.; 81830046 and U1905208 to L.C.; 81902797 to L.H.; 32070891 to J.G.), the Fundamental Research Funds for the Central Universities of China-Xiamen University (20720180047 to L.C. and 20720192009 to D.Z.). "The Young Talent Support Plan" of Xi'an Jiaotong University (to J.G.) and the Thousand Talents Plan of Shaanxi Province (to J.G.). We thank National Center for Protein Sciences at Peking University in Beijing, China, for assistance with Bruker Bio-resolve AFM

and Dr. Chan Li for help with data collection. The funders had no role in study design, data collection and analysis, decision to publish, or preparation of the manuscript.

## Author contributions

D.Z. and L.C. conceived the project with the input from C.W., Y.S., and X.D. J.G., Y.S., J.Z., B.Y., P.W., W.Y., H.Z., J.L., F.Q., L.H., and C.X. performed experimental biological research. D.Z. and L.C. co-wrote the paper. All authors edited the manuscript.

## Competing interests

The authors declare no competing interests.
