## [Peer Review File · Nature Communications]

REVIEWER COMMENTS

Reviewer #1 (Remarks to the Author):

The Authors revealed a mechanism by which mechanical stimuli modulate immune responses of macrophages to LPS and affect E.coli endocytosis. This signaling cascade includes activation of a mechanoreceptor/calcium channel - Piezo1 and the following activity of Mst1/2 kinases of the Hippo pathway, that had been successfully studied by the team earlier. The present study shows that Piezo1 associates with TLR4 in LPS-activated macrophages. The chain of reactions triggered thereby starts from a Piezo1-mediated influx of calcium ions into the cell, the activation of CaMKII kinase and phosphorylation of Mst1/2 kinases, and the activation of Rac1 protein. As a result, actin filaments are reorganized and mitochondria accumulate at phagosomes which is followed by production of ROS with a bactericidal outcome.

The manuscript provides a fairly complete description of this entirely novel signaling pathway induced by LPS which is crucial for combating bacterial infections. This novelty is a great advantage of the study. So far, there were only few papers available concerning the role of Piezo1 in immune reactions; the Piezo1-CaMKII-Mst1/2-Rac1 axis has not been revealed earlier while the study provides a comprehensive picture of its functioning in LPS-stimulated macrophages. The results are solid, the manuscript is well written, methods are described in sufficient detail. The studies are based on several state-of-the-art techniques, including the use of a series of KO and transgenic mice. Taking into account the novelty and quality of the work I recommend the manuscript for publication in Nature Communications after addressing the following minor points:

1. The title should read: TLR4 signaling augments....

Also in some other places TLR should be replaced with TLR4, e.g., page 4 line 85; page 5 line 105; page 11, lines 276, 278, 291.

There are manuscript fragments which refer to studies on TLR2 activity, especially in Methods – application of anti-TLR2 antibody, stimulation of cells with LTA.

The whole manuscript should be carefully checked and amended to focus on TLR4 only as no data on other TLRs are shown.

2. The Authors may refer to paper of Schappe et al, 2018, Immunity 48:59-74.e5, on the involvement of another calcium channel, TRPM7, in LPS-induced signaling of macrophages.

3. Figure miniaturization is going too far, especially curves reflecting FACS results are hard to read.

4. Fig 3b,d, f- application of two-way Anova can provide better insight in the influence of the Piezo1 depletion on cell stiffness both prior to and after stimulation with various ligands/bacteria; similarly, the analysis of co-localization of E.coli phagosomes with mitochondria (Hsp60 labeling) would benefit from such approach. The same remark concerns Figs 4f,h; 5c, 6n.

5. Fig. 3j: The differences in MitoSOX and CellROX production in Piezo1-positive cells before and after stimulation with LPS, E. coli or L. monocytogenes on 35 kPa surface are barely detectable, yet stated in the text (page 7, lines 183-184).

6. Figure 4e is not convincing in showing the difference of F-actin organization in LPS-stimulated cells (middle panel): in all four conditions the actin organization is alike. An analysis of Figs 3a and 3c suggests that LPS and Yoda1 induce filopodia formation in Piezo1-positive cells. If so, the description of Yoda1- or LPS-induced formation of "straight bundles of actin filaments" (page 6, line 158) should be changed accordingly. This, in turn, will facilitate observations of cytoskeleton disturbances caused by the Piezo1 depletion (Fig. 3c, lower panel), Rac expression (Fig.4e) and Mst1/2 deficiency (Fig. 5b). Also, the actin filament reorganization induced upon phagocytosis of E.coli (Fig. 3c upper panel) with an expected accumulation of microfilaments at phagocytic cups and its disturbances after Piezo1 depletion (Fig. 3c, lower panel) etc. should be described more precisely.

7. Fig. 6c: the Authors claim that : The result showed that removal of extracellular calcium

effectively inhibited Yoda1-induced Mob1 phosphorylation (Fig. 6b), and this effect can only be observed in wild-type BMDMs, but not in Piezo1 deficient BMDMs (Fig. 6c). In fact, Fig. 6c shows results of BAPTA application. Correct the text accordingly. Was DMEM without Ca²⁺ commercial?

8. Fig. 6h- the description of this Figure in the text is not sufficient; what is HA-CALM3? Why Mst1 K/R was used in these studies?

9. Having prized the Authors for a well-written manuscript I need to admit that I have trouble understanding the following sentences:

Page 4, lines 89-90 : To study the mechanobiology in macrophages mediated innate immunity, we determined the expression levels of MSICs on macrophages.

Page 7, lines 180-182. Furthermore, CytD treatment that disrupts actin polymerization abrogated ROS production induced by Yoda1-triggered Piezo1 activation phagocytosis and bactericidal activity (Fig. 3h, i).

Page 9, lines 229-230: These results suggested that kinases Mst1/2 might also involve in Piezo1-mediated Rac1 activation and cytoskeleton reorganization.

Pages 9/10 lines 245-255:indicating that TLR4 signaling induces calcium signals might partially through the activation of the ion channel Piezo1 (Fig. 6a).

Page 36, description of Fig. 4b is not clear.

10. Other flaws:

- Listeria monocytogenes not Listeria Monocytogenes;
 - The catalog number of the anti-TLR4 antibody is wrong;
 - What is sfGFP-TLR4 construct?
 - In which experiments was the liver tissue stained with anti-beta-catenin? (page 20, line 528).
- Take care of the Methods section to include only information relevant to the study.

Reviewer #2 (Remarks to the Author):

In this manuscript, the authors report that TLR4 stimulates a Piezo1 mediated mechanism of bacterial killing. Unfortunately, the data presented is not supportive of a compelling biological model. While I agree that Piezo1 influences cytoskeletal activities in some manner. The interactions with TLR4 are not compelling and the functional activities described are very weak. Two major comments are indicated below, although these are not the only concerns I have with this study.

1. The studies in Figure 1 describe intracellular staining for TLR4. This protein has been notoriously difficult to visualize reliably. As such, I am surprised by the quality of the data obtained here. The authors must perform staining of TLR4 KO BMDMs to verify specificity. A similar concern is offered for the westerns for TLR4 in this Figure. KO studies must be performed. Additionally, no negative controls are offered in this Figure. Does TLR2 or TLR9 behave similarly?

2. The phenotypes of bacterial killing associated with Piezo1 KOs in Figure 2 are so modest that the significance of the findings are questionable. Similar concerns associated with the data in Figure 4. Bacterial killing, which is highlighted by the authors as a critical function, is very modest. For example, examine Figure 4D.

Point-by-point response to reviewers

TLR4 signaling augments macrophage bactericidal activity through Piezo1

(Manuscript # NCOMMS-20-23301)

Reply to Reviewer 1:

We would like to thank the reviewer for the positive comments on our manuscript. In response to these comments, we now have revised manuscript with some new data. Please note that all changes in the manuscript are highlighted with yellow.

Reviewer #1 (Remarks to the Author):

The Authors revealed a mechanism by which mechanical stimuli modulate immune responses of macrophages to LPS and affect E.coli endocytosis. This signaling cascade includes activation of a mechanoreceptor/calcium channel - Piezo1 and the following activity of Mst1/2 kinases of the Hippo pathway, that had been successfully studied by the team earlier. The present study shows that Piezo1 associates with TLR4 in LPS-activated macrophages. The chain of reactions triggered thereby starts from a Piezo1-mediated influx of calcium ions into the cell, the activation of CaMKII kinase and phosphorylation of Mst1/2 kinases, and the activation of Rac1 protein. As a result, actin filaments are reorganized and mitochondria accumulate at phagosomes which is followed by production of ROS with a bactericidal outcome.

The manuscript provides a fairly complete description of this entirely novel signaling pathway induced by LPS which is crucial for combating bacterial infections. This novelty is a great advantage of the study. So far, there were only few papers available concerning the role of Piezo1 in immune reactions; the Piezo1-CaMKII-Mst1/2-Rac1 axis has not been revealed earlier while the study provides a comprehensive picture of its functioning in LPS-stimulated macrophages. The results are solid, the manuscript is well written, methods are described in sufficient detail. The studies are based on several state-of-the-art techniques, including the use of a series of KO and transgenic mice. Taking into account the novelty and quality of the work I recommend the manuscript for publication in Nature Communications after addressing the following minor points:

Answer: We thank the reviewer for the positive comments on our manuscript. We acknowledge and appreciate the critical issues raised by the reviewer.

1. The title should read: TLR4 signaling augments....

Also in some other places TLR should be replaced with TLR4, e.g., page 4 line 85; page 5 lane 105; page 11, lines 276, 278, 291.

There are manuscript fragments which refer to studies on TLR2 activity, especially in Methods – application of anti-TLR2 antibody, stimulation of cells with LTA.

The whole manuscript should be carefully checked and amended to focus on TLR4 only as no data on other TLRs are shown.

Answer: We apologize that we did not make precise presentation in the manuscript. We have carefully checked the manuscript and made the change according to the reviewer's suggestion. In addition, the manuscript title has been changed to "TLR4 signaling augments macrophage bactericidal activity through Piezo1". We specifically state the "TLR4" instead of the "TLR" in the updated manuscript. Some sentences related to TLR2 study have been deleted.

2. The Authors may refer to paper of Schappe et al, 2018, Immunity 48:59-74.e5, on the involvement of another calcium channel, TRPM7, in LPS-induced signaling of macrophages.

Answer: We have added this reference in the page 9 of the updated manuscript.

3. Figure miniaturization is going too far, especially curves reflecting FACS results are hard to read.

Answer: To address the reviewer's concern, we have enlarged the FACS images in the updated manuscript. In addition, we inserted the enlarged details of the filopodia formation or actin filament reorganization for all F-actin immunofluorescence staining figures.

4. Fig 3b,d, f- application of two-way Anova can provide better insight in the influence of the Piezo1 depletion on cell stiffness both prior to and after stimulation with various ligands/bacteria; similarly, the analysis of co-localization of E.coli phagosomes with mitochondria (Hsp60 labeling) would benefit from such approach. The same remark concerns Figs 4f,h; 5c, 6n.

Answer: Following the reviewer's insightful suggestion, we reanalyze the relevant data following two-way ANOVA and present the results in the updated manuscript.

5. Fig. 3j: The differences in MitoSOX and CellROX production in Piezo1-positive cells before and after stimulation with LPS, E. coli or L. monocytogenes on 35 kPa surface are barely detectable, yet stated in the text (page 7, lines 183-184).

Answer: We apologize to the reviewer that we did not present the data in a clear manner. We have stated this observation in the updated manuscript. On page 7 "In contrast, the mROS or cellular ROS could be induced in *Piezo1*^{fl/fl} BMDMs cultured on 35 kPa matrix with a tiny amount production, or on glass (much stiffer matrix) with much more ROS production upon various stimulations, but ROS induction were diminished in *Piezo1*^{fl/fl} *Lyz2-Cre* BMDMs cultured on any types of matrix (Fig. 3j, Supplementary Figure 3b)."

6. Figure 4e is not convincing in showing the difference of F-actin organization in LPS-stimulated cells (middle panel): in all four conditions the actin organization is alike.

An analysis of Figs 3a and 3c suggests that LPS and Yoda1 induce filopodia formation in Piezo1-positive cells. If so, the description of Yoda1- or LPS-induced formation of "straight

bundles of actin filaments” (page 6, line 158) should be changed accordingly. This, in turn, will facilitate observations of cytoskeleton disturbances caused by the Piezo1 depletion (Fig. 3c, lower panel), Rac expression (Fig.4e) and Mst1/2 deficiency (Fig. 5b). Also, the actin filament reorganization induced upon phagocytosis of E.coli (Fig. 3c upper panel) with an expected accumulation of microfilaments at phagocytic cups and its disturbances after Piezo1 depletion (Fig. 3c, lower panel) etc. should be described more precisely.

Answer: Following the reviewer’s insightful suggestion, the description of “straight bundles of actin filaments” was replaced with the “filopodia formation” in the updated manuscript.

In the middle panel of Figure 4e, Piezo1 deficient macrophage treated with LPS has much fewer and shorter filopodia, while reintroduction of Rac1^{G12V} enhanced and restored the filopodia formation in *Piezo1^{fl/fl}* and *Piezo1^{fl/fl} Lyz2-Cre* macrophages respectively. In addition, we have inserted the enlarged images of the filopodia formation or actin filament reorganization for all F-actin immunofluorescence staining figures (Fig. R1, Figure 6j in the updated manuscript). In addition, we added the enlarged details of the filopodia formation or actin filament reorganization for all F-actin immunofluorescence staining figures.

Figure R1. Confocal microscopy of *Piezo1^{fl/fl}* or *Piezo1^{fl/fl} Lyz2-Cre*, *Rac1^{G12V} Lyz2-Cre* or *Piezo1^{fl/fl} Rac1^{G12V} Lyz2-Cre* BMDMs treated with control PBS, LPS (1 $\mu\text{g ml}^{-1}$) or *E.coli* (MOI, 20) for 30 min, then immunostained with anti-F-actin (green) and counterstained with DAPI (blue); outlined areas are enlarged in inserts in corners. Scale bars, 20 μm .

7. Fig. 6c: the Authors claim that The result showed that removal of extracellular calcium effectively inhibited Yoda1-induced Mob1 phosphorylation (Fig. 6b), and this effect can only be observed in wild-type BMDMs, but not in Piezo1 deficient BMDMs (Fig. 6c). In fact, Fig. 6c

shows results of BAPTA application. Correct the text accordingly. Was DMEM without Ca²⁺ commercial?

Answer: We thank the reviewer for the constructive comments. We corrected the text to “The result showed that removal of extracellular calcium effectively inhibited Yoda1-induced Mst1/2 activation, shown as diminished Mob1 phosphorylation (Fig. 6b). Consistently, elimination of intracellular calcium by calcium chelator BAPTA-AM treatment also effectively disrupted the Yoda1-induced Mst1/2 activation in a time-dependent manner and a dose-dependent manner (Fig. 6c-e), and this effect can only be observed in wild-type BMDMs, but not in Piezo1 deficient BMDMs (Fig. 6c).” on the page 10. The DMEM without Ca²⁺ is commercial available (Gibco/Thermofisher, Cat:21068028).

8. Fig. 6h- the description of this Figure in the text is not sufficient; what is HA-CALM3? Why Mst1 K/R was used in these studies?

Answer: We apologize to the reviewer that we did not describe the data in a clear manner. We have rewritten the text in the updated manuscript. “In addition, the phosphorylation of Mst1K/R, which cannot be auto-phosphorylated, was observed in the presence of CaMKII, suggesting Mst1 is a substrate of CaMKII (Fig. 6h).” on the page 10 of the updated manuscript.

The HA-CALMs is the HA-tagged Calmodulin-3, a Ca²⁺-binding protein that transduces Ca²⁺-mediated signals by binding to and enhancing the CaMKII activity. To determine whether Mst1 is a substrate of CaMKII kinase and avoid the auto-phosphorylation effect of Mst1, Mst1K/R, a kinase-dead form of Mst1, in which the K (Lysine) is replaced with R (Arginine), was used in the Phos-tag assay.

9. Having prized the Authors for a well-written manuscript I need to admit that I have trouble understanding the following sentences:

Answer: We apologize to the reviewer that we did not make the presentation in a clear manner. We have rewritten these sentences.

Page 4, lines 89-90 : To study the mechanobiology in macrophages mediated innate immunity, we determined the expression levels of MSICs on macrophages.

Answer: This sentence has been revised to “To study the function of mechanosensitive ion channels (MSICs) in innate immunity, we determined the expression levels of MSICs on macrophages.”

Page 7, lines 180-182. Furthermore, CytD treatment that disrupts actin polymerization abrogated ROS production induced by Yoda1-triggered Piezo1 activation phagocytosis and bactericidal activity (Fig. 3h, i).

Answer: This sentence has been revised to “Furthermore, CytD treatment that disrupts actin polymerization abrogated ROS production (Fig. 3h), and diminished the Yoda1-induced phagocytosis and bactericidal activity in BMDMs (Fig. 3h, i).”

Page 9, lines 229-230: These results suggested that kinases Mst1/2 might also involve in Piezo1-mediated Rac1 activation and cytoskeleton reorganization.

Answer: This sentence has been revised to “These results suggested that kinases Mst1/2 might be involved in the regulation of Piezo1-mediated Rac1 activation and filopodia formation.”

Pages 9/10 lines 245-255:indicating that TLR4 signaling induces calcium signals might partially through the activation of the ion channel Piezo1 (Fig. 6a).

Answer: This sentence has been revised to “Interestingly, the LPS-induced calcium influx was also dramatically suppressed in Piezo1-deficient BMDMs, indicating that the ion channel Piezo1 is important for the induction of calcium influx by TLR4 signaling (Fig. 6a).” on the page 9.

Page (8) 36, description of Fig. 4b is not clear.

Answer: The description of Fig. 4b has been revised to “Although the abundance of total Rac1 in Piezo1-deficient BMDMs was similar to that in wild-type BMDMs, the amount of active form of Rac1 (i.e. Rac-GTP) was much lower in Piezo1-deficient BMDMs than in wild-type BMDMs (Fig. 4b). Yoda1 treatment further increased the amount of Rac1-GTP in wild-type but not in Piezo1-deficient BMDMs (Fig. 4b).” on the page 8.

10. Other flaws:

Answer: We apologize to the reviewer for those oversights and errors.

-Listeria monocytogenes not Listeria Monocytogenes:

Answer: This error has been corrected in the updated manuscript.

-The catalog number of the anti-TLR4 antibody is wrong:

Answer: The anti-TLR4 antibody used in this study was obtained from Proteintech (19811-1-AP). We have corrected this mistake in the updated manuscript.

- What is sfGFP-TLR4 construct?

Answer: The construct used in this study is GFP-TLR4. We have corrected this mistake in the updated manuscript.

- In which experiments was the liver tissue stained with anti-beta-catenin? (page 20, line 528).

Answer: We apologize to the reviewer for this oversight. We have deleted this description in the Method section.

Take care of the Methods section to include only information relevant to the study.

Answer: We thank for the reviewer’s comment. We have carefully checked the Methods section and corrected all the errors.

Reply to Reviewer 2:

We thank the reviewer for the positive comments on our manuscript. We acknowledge and appreciate the critical issues raised by the reviewer. We have added a significant amount of data in the revised manuscript according to the reviewer's constructive comments and all changes in the updated manuscript are highlighted with yellow.

Reviewer #2 (Remarks to the Author):

In this manuscript, the authors report that TLR4 stimulates a Piezo1 mediated mechanism of bacterial killing. Unfortunately, the data presented is not supportive of a compelling biological model. While I agree that Piezo1 influences cytoskeletal activities in some manner. The interactions with TLR4 are not compelling and the functional activities described are very weak. Two major comments are indicated below, although these are not the only concerns I have with this study.

1. The studies in Figure 1 describe intracellular staining for TLR4. This protein has been notoriously difficult to visualize reliably. As such, I am surprised by the quality of the data obtained here. The authors must perform staining of TLR4 KO BMDMs to verify specificity. A similar concern is offered for the westerns for TLR4 in this Figure. KO studies must be performed. Additionally, no negative controls are offered in this Figure.

Answer: We agree with the reviewer's concern. Following the reviewer's insightful suggestion, we verified the specificity of anti-TLR4 antibody (Proteintech, 19811-1-AP) by performing the TLR4 immunofluorescence staining on mixed cells composed of unlabeled wild-type (WT) BMDMs and CFSE-labeled TLR4-deficient BMDMs, and vice versa, as indicated in Figure R2A. The result clearly showed that the fluorescent signals with anti-TLR4 antibody were dramatically diminished in TLR4 KO BMDMs as compared with WT cells (Figure R2A). In addition, an immune blotting assay performed with anti-TLR4 antibody also exhibited much clearer and stronger TLR4 bands in WT BMDMs than in TLR4 KO BMDMs (Figure R2B). In the manuscript, we demonstrated that the LPS treatment resulted in the co-localization of Piezo1 and TLR4 receptors in BMDMs by using a structure illumination microscopy (SIM) approach (Figure R2C, or Fig. 1d in the manuscript). To further verify the specificity of TLR4 antibody, the WT and TLR4 KO BMDMs treated with LPS or infected with DAPI-labeled *E.coli* were stained with anti-TLR4 antibody and Alexa Fluor 555-conjugated donkey anti-rabbit IgG(H+L) antibody (A31572, Invitrogen), and the fluorescent images were collected with a Precision DeltaVision-OMX Super-Resolution Microscope (GE OMX V4). TLR4 expression was clearly detected in the WT but not in TLR4-deficient BMDMs (Figure R2D and E). The accumulation of TLR4 receptors or co-localization of TLR4 with *E.coli* can only be observed in WT but not TLR4 KO BMDMs upon LPS stimulation or *E. coli* infection respectively (Figure R2D and E). Moreover, in the

Figure 1E in the manuscript, we performed immunoprecipitation assays with anti-RFP antibody to precipitate RFP-tagged Piezo1 and IgG antibody as the negative control. We did not detect the TLR4 molecule in the negative control precipitated sample, suggesting that TLR4 is specifically bound to Piezo1. In sum, these result suggested that the anti-TLR4 antibody (Proteintech, 19811-1-AP) is quite specific for labeling the TLR4 receptor and TLR4 is specifically bound to Piezo1.

Figure R2. (A) Fluorescent microscopy of TLR4 (red) in WT BMDMs (left) or TLR4 KO BMDMs (right) labeled with CFSE (green), then mixed with unlabeled TLR4 KO cells (stars) or wild-type BMDMs (arrows). Scale bars, 20 μ m. (B) Immunoblot analysis of TLR4, β -actin and GAPDH in WT or TLR4 KO BMDMs stimulated with or without LPS (1 μ g ml⁻¹). (C) SIM of the co-localization of Piezo1 (red) and TLR4 (green) in *Piezo1*^{P1tdT} BMDMs treated with or without LPS (1 μ g ml⁻¹) for 30 min; $\times 16$ magnification of areas outlined in the main images are shown next to the main images. Scale bars, 20 μ m. (D) SIM of the TLR4 (red) in WT and TLR4 KO BMDMs treated with or without LPS (1 μ g ml⁻¹) for 30 min; $\times 16$ magnification of areas outlined in the main images are shown next to the main images. Scale bars, 20 μ m. (E) SIM of the co-localization of TLR4 (red) and *E. coli* (blue) in WT and TLR4 KO BMDMs infected with DAPI labeled *E. coli*; $\times 16$ magnification of areas outlined in the main images are shown next to the main images. Scale bars, 20 μ m.

Does TLR2 or TLR9 behave similarly?

Answer: Interestingly, we found that TLR2 behaves similarly to TLR4, but TLR9 does not. Our previous study showed that the stimulation of cell-surface TLRs (TLR1, TLR2 and TLR4) substantially enhanced the phosphorylation of Mob1, a physiological substrate of the kinases Mst1/2, whereas the ligands of endosomal TLRs (R848 for TLR7 and TLR8 or CpG DNA for TLR9) did not change the phosphorylation of Mob1 (Geng J et al Nature Immunology, 2015, Figure R3A). Furthermore, engagement of the cell-surface TLRs (TLR1, TLR2 and TLR4) augmented mROS and total cellular ROS in BMDMs but engagement of the endosomal TLRs (TLR3, TLR7, TLR8 and TLR9) did not (Geng J et al Nature Immunology, 2015, Figure R3B). These results suggested that cell-surface TLRs, such as TLR2 and TLR4 and endosomal TLRs, such as TLR9 have different effects on Hippo signaling activation.

A (adapted from Fig. 1f and Suppl. Fig. 1i, Geng J. et al, Nature Immuno, 2015)

B (adapted from Fig. 2g and Suppl. Fig. 2a, Geng J. et al, Nature Immuno, 2015)

Figure R3. (A) Immunoblot analysis of phosphorylated (p-) Mob1 p38 and Jnk, and total IκBα and GAPDH in wild-type BMDMs stimulated for 0–60 min with LPS, LTA or CpG. (B) Flow cytometry analyzing mROS production by unstimulated wild-type BMDMs left unstained or by wild-type and Mst1/2 DKO BMDMs left unstimulated (ROS dye alone) or stimulated for 6 h with the TLR agonist LPS, Pam3CSK4 (Pam3), LTA, CpG, PIC or R848 (ROS dye + agonist), then stained for 30 min with MitoSOX.

Here, we found that LTA (an agonist of TLR2) stimulation induced the co-localization of TLR2 receptor with Piezo1 (Figure R4A and B), while CpG (an agonist of TLR9) treatment didn't trigger the formation of the TLR9 and Piezo1 complex in BMDMs (Figure R4C). In addition, LTA but not CpG treatment resulted in filopodia formation, enhanced cellular stiffness and ROS induction in WT but not Piezo1 KO BMDMs (Figure R4D-G). These phenotypes are similar to that with LPS, *E. coli* or Yoda1 treatment. Consistent with the LPS, *E. coli* or *Listeria* treatments, LTA-induced ROS production also mainly depends on the

matrix stiffness and Piezo1 receptors (Figure R4H). Similarly, LTA treatment increased the phosphorylation levels of Mob1, a physiological substrate of the kinases Mst1/2 in wild-type BMDMs, but not in Piezo1 deficient BMDMs, suggesting the TLR2 signaling might activate Mst1/2 signaling through Piezo1 (Figure R4I). These results suggested that cell-surface TLRs, such as TLR2 and TLR4 are involved in Piezo1-mediated Mst1/2 activation for ROS production, but endosomal TLRs, such as TLR9, do not

Figure R4. (A) SIM of *Piezo1*^{P1tdT} BMDMs incubated with uncoated or LTA-coated latex beads, followed by immunostaining of Piezo1 or TLR2 as indicated; white arrows indicate colocalization of beads (purple), TLR4 (green) and Piezo1 (red), Scale bars, 20 μ m. (B and C) SIM of the Piezo1 (red) and TLR2 (green) or TLR9 (green) in *Piezo1*^{P1tdT} BMDMs treated with or without LTA or CpG as indicated for 30 min; $\times 16$ magnification of areas outlined in the main images are shown next to the main images. Scale bars, 20 μ m. (D) Confocal microscopy of *Piezo1*^{fl/fl} or *Piezo1*^{fl/fl} *Lyz2*-Cre BMDMs treated with LPS (1 μ g ml⁻¹), LTA (1 μ g ml⁻¹), CpG (50 μ M) *E. coli* (MOI, 20), Yoda1 (5 μ M), then immunostained with anti-F-actin (green) and counterstained with DAPI (blue). Scale bars, 20 μ m. (E and F) The Young's modulus was determined by AFM for *Piezo1*^{fl/fl} (E) or *Piezo1*^{fl/fl} *Lyz2*-Cre (F) BMDMs treated with PBS control (n=104, 98), LTA (1 μ g ml⁻¹, n=101,102), CpG (50 μ M, n=100, 100), LPS (1 μ g ml⁻¹, n=55, 105), *E. coli* (n=95, 93) or *L. monocytogenes* (LM, n=91,91) for 30 min. The whiskers show the 5–95th percentiles. ns, not significant (P > 0.05); *P < 0.05, **P < 0.01, ***P < 0.001, ****P < 0.0001 as indicated by bracketing, Two-way ANOVA (G) Flow cytometry analyzing mROS and cellular ROS production by *Piezo1*^{fl/fl} or *Piezo1*^{fl/fl} *Lyz2*-Cre BMDMs stimulated with LTA (1 μ g ml⁻¹) or CpG (50 μ M) for 3h followed by staining with MitoSOX or CellROX. (H) Flow cytometry analyzing mROS and cellular ROS production by *Piezo1*^{fl/fl} or *Piezo1*^{fl/fl} *Lyz2*-Cre BMDMs treated LPS (1 μ g ml⁻¹) for 3 h, *E. coli* (MOI, 20) or *L. monocytogenes* (MOI, 10) for 1 h on different stiffness culture matrix of 5kPa, 35 kPa, or glass, followed by staining with MitoSOX or CellROX.

2. The phenotypes of bacterial killing associated with Piezo1 KO in Figure 2 are so modest that the significance of the findings are questionable. Similar concerns associated with the data in Figure 4. Bacterial killing, which is highlighted by the authors as a critical function, is very modest. For example, examine Figure 4D.

Answer: Numerous studies have shown that the Toll-like receptor (TLR) family plays a fundamental role in pathogen recognition and activation of innate immunity. Although many reported molecules, ranging from membrane and cytosol to nuclear, contribute to TLR ligand discrimination or receptor signaling via different mechanisms, our current study clearly demonstrated that Piezo1 played an important role in innate immunity. For example, the bacterial peritonitis or *Listeria monocytogenes* infection killed ~80% of *Piezo1*^{fl/fl}*Lyz2*-Cre mice but only ~40% of *Piezo1*^{fl/fl} littermates (Fig. 2a in the updated manuscript) *in vivo*. *In vitro* bacterial killing assays also showed that BMDMs cultured on soft matrix exhibited significantly lower bactericidal activity than that on stiffer matrix after *E. coli* infection (Fig. 2e in the updated manuscript), while depletion of Piezo1 abolished the bactericidal activity of BMDMs cultured on the stiffer matrix (Figure 2f in the updated manuscript). These results indicated that both the proper extracellular matrix stiffness and Piezo1 receptor are important for macrophage phagocytosis and bactericidal activity. The stiffer microenvironments of acute

and chronic inflammation that macrophages are often present in highlights the necessity to consider mechanical sensing signals that may influence macrophage functions. In sum, our current study provides insight into the mechanism and function of Piezo1 governing mechanical transduction system during the infection.

REVIEWER COMMENTS

Reviewer #1 (Remarks to the Author):

I recommend the manuscript for publication after addressing two points:

1. Statistics. The Authors performed two-way Anova test on most data indicated. In Fig. 4 the "k" panel mentioned in the Figure legend refers probably to "j" panel.

Statement that "The two-way Anova analysis was used for comparing cell stiffness or the colocalization of E. coli phagosomes with mitochondria between two groups" is not clear. Anova with a post-test should be used for multiple comparisons in one graph, instead of many pairs analyzed separately with Student's t test. In addition, the post-test used is not indicated. Please, look carefully at your data and apply the Anova test with an appropriate post-hoc test, and describe the test accordingly.

2. Indicate the catalog number of LPS.

Reviewer #2 (Remarks to the Author):

In this revised manuscript, the authors have attempted to address my prior concerns. Unfortunately, those concerns remain. The new microscopy data offered is blurry and not consistent with the staining expected of a plasma membrane localized protein.

The reply to the weak bacterial killing phenotypes is also not compelling. The authors cannot deny that their bacterial killing data in Figure 2E and F (referenced in their rebuttal) is weak. 2 fold differences (or less) in CFU counts minimize the impact of this work.

Point-by-point response to reviewers

TLR4 signaling augments macrophage bactericidal activity through Piezo1

(Manuscript # NCOMMS-20-23301A)

Reviewer #1 (Remarks to the Author):

I recommend the manuscript for publication after addressing two points:

1. Statistics. The Authors performed two-way Anova test on most data indicated. In Fig. 4 (should be the Fig. 6) the “k” panel mentioned in the Figure legend refers probably to “j” panel.

Statement that “The two-way Anova analysis was used for comparing cell stiffness or the colocalization of E. coli phagosomes with mitochondria between two groups” is not clear. Anova with a post-test should be used for multiple comparisons in one graph, instead of many pairs analyzed separately with Student’s t test. In addition, the post-test used is not indicated. Please, look carefully at your data and apply the Anova test with an appropriate post-hoc test, and describe the test accordingly.

2. Indicate the catalog number of LPS.

We thank the reviewer for the positive comments and suggestions. We have accordingly revised the manuscript carefully to better convey our views.

- 1) We have corrected the mislabeling of “k” to “j” in Fig. 6 legend. We apologized for not stating the two-way Anova analysis in methods correctly. We have checked carefully at our data and make sure that the two-way Anova analysis were done to compare multiple groups in one graph.
- 2) The catalog number of LPS is Sigma L4391. We have updated it in the manuscript.

Reviewer #2 (Remarks to the Author):

In this revised manuscript, the authors have attempted to address my prior concerns. Unfortunately, those concerns remain. The new microscopy data offered is blurry and not consistent with the staining expected of a plasma membrane localized protein.

We appreciated the critical issues raised by the reviewer. To our knowledge, TLR4 is unique among pathogen-recognition receptors in that it initiates different pathways in different cellular locations. Piezo1 is a large multipass transmembrane protein which acts at both the plasma membrane and in intracellular compartments. Per the reviewer's request, we further performed TLR4, F4/80 (a plasma membrane protein) and Piezo1 co-staining in BMDMs from the Piezo1^{P1tdT} transgenic mice, collected the images by confocal microscope. We observed that some of TLR4 were indeed aligned with F4/80 and localized with plasma membrane while LPS treatment can trigger the association of TLR4 with Piezo1 to form condensed spots in intracellular compartments (Fig. R1A). In addition, we performed immunofluorescence staining with TLR4 and F4/80 in BMDMs from wild-type (WT) or TLR4 KO mice and collected the images with the confocal microscope. We observed that some of TLR4 were aligned with F4/80 in WT BMDMs, but TLR4 could not be detected in TLR4 KO BMDMs indicating that the antibody against TLR4 is specific (Fig. R1B). Furthermore, BMDMs were incubated with uncoated or LPS-coated latex beads to elicit phagocytosis of these beads. The result clearly demonstrated that TLR4 was localized with membrane wrapped on the LPS-coated beads and much higher amount of Piezo1 was recruited and co-localized with TLR4 on the LPS-coated beads than that on the uncoated beads, as observed with a structure illumination microscopy (SIM) approach (Fig. R1C).

The reply to the weak bacterial killing phenotypes is also not compelling. The authors cannot deny that their bacterial killing data in Figure 2E and F (referenced in their rebuttal) is weak. 2 fold differences (or less) in CFU counts minimize the impact of this work.

While we appreciated the reviewer's concern, we respectfully disagree. Our results clearly showed that the bacterial peritonitis or *Listeria monocytogenes* (LM) infection killed ~80% of Piezo1^{fl/fl}Lyz2-Cre mice but only ~40% of Piezo1^{fl/fl} littermates (Figure 2a, in the updated manuscript) *in vivo*. *In vitro* bacterial killing assays showed that BMDMs cultured on soft matrix exhibited significantly lower bactericidal activity than that on stiffer matrix after *E. coli* infection (Figure 2e, in the updated manuscript) and lack of Piezo1 abolished the enhanced bactericidal activity on the stiffer matrix (Figure 2f, in the updated manuscript). These results indicated that the proper matrix stiffness and Piezo1 receptor is important for macrophage phagocytosis and bactericidal activity. Mechanistically, our further study showed that LPS stimulates TLR4 to induce Piezo1-mediated calcium influx and consequently activates CaMKII-Mst1/2-Rac axis for pathogen ingestion and killing. The stiffer microenvironments of acute and chronic inflammation that macrophages are often present in suggested that mechanical sensing signals may be one of the regulating signals for TLR signaling to influence macrophage functions.

Fig.R1 (A) Confocal immunostaining of the Piezo1 (red), TLR4 (green), F4/80 (purple) and DAPI (blue) in *Piezo1*^{P1tdT} BMDMs treated with or without LPS. (B) Confocal immunostaining of TLR4 (red), F4/80 (green) and DAPI (blue) in WT or TLR4 KO BMDMs treated with or without LPS. (C) SIM of *Piezo1*^{P1tdT} BMDMs incubated with uncoated or LPS-coated latex beads, followed by immunostaining of Piezo1 or TLR4 as indicated; white arrows indicate colocalization of beads (purple), TLR4 (green) and Piezo1 (red). Scale bars, 20 μ m.

REVIEWER COMMENTS

Reviewer #2 (Remarks to the Author):

In this revised manuscript, the authors have attempted to address my prior concerns. New microscopy data is offered that seems to suggest that TLR4 staining is accurate, based on the use of TLR4 KO controls. This staining is not ideal though, as they did not use TLR4 KO cells in most assays for microscopy. The most critical experiments that would have benefitted from this control are those examining TLR4 localization to beads, where the authors observe impressive colocalization of beads with TLR4. If that controls are offered, this reviewer would be satisfied that the staining is accurate.

Most importantly is the functional conclusions offered in this study on bacterial killing. The authors do not rebut my concern that bacterial killing is very modest in this study. Indeed, less than a 2 fold difference in bacterial CFUs are observed in most analyses. Moreover, the authors have a similar in vivo phenotype when examining E.coli and Listeria. The latter is Gram positive bacterium that does not contain LPS. As such, it is not clear to me that the in vivo phenotypes observed are related to LPS signaling phenotypes.

While I appreciate the quality of select sections of the data presented, the importance of the findings offered here remain in question.

Point-by-point response to reviewers

TLR4 signaling augments macrophage bactericidal activity through Piezo1

(Manuscript # NCOMMS-20-23301B)

Reviewer #2 (Remarks to the Author):

In this revised manuscript, the authors have attempted to address my prior concerns. New microscopy data is offered that seems to suggest that TLR4 staining is accurate, based on the use of TLR4 KO controls. This staining is not ideal though, as they did not use TLR4 KO cells in most assays for microscopy. The most critical experiments that would have benefitted from this control are those examining TLR4 localization to beads, where the authors observe impressive colocalization of beads with TLR4. If that controls are offered, this reviewer would be satisfied that the staining is accurate.

Answer: Per the reviewer's advice, we have shown in the previously revised manuscript, that, with wild-type or TLR4 KO macrophages, TLR4 was indeed localized with plasma membrane and LPS treatment can trigger the association of TLR4 with Piezo1 to form condensed spots in intracellular compartments. To further address the reviewer's concern, we isolated BMDMs from *Piezo1*^{P1tdT} or *Tlr4*^{-/-}*Piezo1*^{P1tdT} mice in which a Piezo1-tdTomato fusion protein is expressed with or without the deletion of TLR4 receptor. We then performed phagocytosis experiment with LPS-coated or uncoated latex beads. The result clearly demonstrated that TLR4 was localized with membrane wrapped on the LPS-coated beads and much higher amount of Piezo1 was recruited and co-localized with TLR4 on the LPS-coated beads than that on the uncoated beads, while none of TLR4 can be detected on LPS-coated beads in *Tlr4*^{-/-}*Piezo1*^{P1tdT} BMDMs (Fig. R1). We have replaced the Figure 1c with new data in the updated manuscript.

Fig.R1 (A) Confocal immunostaining of *Piezo1*^{P1tdT} or *Tlr4*^{-/-}*Piezo1*^{P1tdT} BMDMs incubated with uncoated or LPS-coated latex beads, followed by immunostaining of Piezo1 or TLR4 as indicated; white arrows indicate colocalization of beads (purple), TLR4 (green) and Piezo1 (red). Scale bars, 20 μ m.

Most importantly is the functional conclusions offered in this study on bacterial killing. The authors do not rebut my concern that bacterial killing is very modest in this study. Indeed, less than a 2 fold difference in bacterial CFUs are observed in most analyses. Moreover, the authors have a similar in vivo phenotype when examining E.coli and Listeria. The latter is Gram positive bacterium that does not contain LPS. As such, it is not clear to me that the in vivo phenotypes observed are related to LPS signaling phenotypes.

Answer: Thanks for the note. Our results clearly showed that the bacterial peritonitis killed ~80% of *Piezo1*^{fl/fl}*Ly2-Cre* mice but only ~40% of *Piezo1*^{fl/fl} littermates (Figure 2a, in the updated manuscript) *in vivo*. *In vitro* bacterial killing assays showed that BMDMs cultured on soft matrix exhibited significantly lower bactericidal activity than that on stiffer matrix after *E. coli* infection (Figure 2e, in the updated manuscript) and lack of Piezo1 abolished the enhanced bactericidal activity on the stiffer matrix (Figure 2f, in the updated manuscript). Mechanistically, our further study showed that LPS stimulates TLR4 to induce Piezo1-mediated calcium influx and consequently activates CaMKII-Mst1/2-Rac axis for pathogen ingestion and killing. These results indicated that the proper matrix stiffness and Piezo1 receptor is important for macrophage phagocytosis and bactericidal activity. Several studies have shown that the TLR family plays a fundamental role in pathogen recognition and activation of innate immunity. Many molecules, ranging from membrane and cytosol to nuclear, contribute to TLR ligand discrimination or receptor signaling and play different roles in the regulation of TLR responses via different mechanisms. The stiffer microenvironments of acute and chronic inflammation that macrophages are often present in. Our study suggested that mechanical sensing signals may be another important mechanism responsible for TLR signals-mediated immune response in macrophages.

We did observe a similar *in vivo* phenotype when examining *E. coli* and *Listeria* infections. As shown in the first rebuttal letter, we found that LTA, an agonist of TLR2 which recognizes a variety of microbial components derived from Gram-positive bacteria, such as *Listeria*, induced the co-localization of TLR2 receptor with Piezo1 (Fig. R2A and B). In addition, LTA treatment also resulted in filopodia formation, enhanced cellular stiffness and ROS induction in WT but not Piezo1 KO BMDMs (Fig. R2C). These phenotypes are similar to that with LPS, *E. coli* or Yoda1 treatment. Consistent with the LPS, *E. coli* treatments, LTA or *Listeria* induced ROS production also mainly depends on Piezo1 receptors (Fig. R2D) and the matrix stiffness (Fig. R2E). Similarly, LTA treatment increased the phosphorylation levels of Mob1, a physiological substrate of the kinases Mst1/2 in wild-type BMDMs, but not in Piezo1 deficient BMDMs, suggesting the TLR2 signaling might activate Mst1/2 signaling through Piezo1 (Fig. R2F). These results suggested that cell-surface TLRs, such as TLR2 and TLR4 are involved in Piezo1-mediated Mst1/2 activation for ROS production and bacterial killing. These observations also suggested that Piezo1 can act as a co-receptor of different types of TLRs, such as TLR4 and TLR2, to translate extracellular chemical signals into mechanical

signals inside the innate cell. However, to focus on the role of TLR4-Piezo1 signaling in macrophage, we didn't put the TLR2 data in this manuscript. We will remove the LM infection data to avoid confusion.

Fig. R2. (A) SIM of *Piezo1*^{P1tdT} BMDMs incubated with uncoated or LTA-coated latex beads, followed by immunostaining of Piezo1 or TLR2 as indicated; white arrows indicate colocalization of beads (purple), TLR2 (green) and Piezo1 (red), Scale bars, 20 μ m. (B) SIM of the Piezo1 (red) and TLR2 (green) in *Piezo1*^{P1tdT} BMDMs treated with or without LTA as indicated for 30 min; $\times 16$ magnification of areas outlined in the main images are shown next to the main images. Scale bars, 20 μ m. (C) Confocal microscopy of *Piezo1*^{fl/fl} or *Piezo1*^{fl/fl} *Lyz2-Cre* BMDMs treated with LPS (1 μ g ml⁻¹), LTA (1 μ g ml⁻¹), *E.coli* (MOI, 20), Yoda1 (5 μ M), then immunostained with anti-F-actin (green) and counterstained with DAPI (blue). Scale bars, 20 μ m. (D) Flow cytometry analyzing mROS and cellular ROS production by *Piezo1*^{fl/fl} or *Piezo1*^{fl/fl} *Lyz2-Cre* BMDMs stimulated with LTA (1 μ g ml⁻¹) for 3h followed by staining with MitoSOX or CellROX. (E) Flow cytometry analyzing mROS and cellular ROS production by *Piezo1*^{fl/fl} or *Piezo1*^{fl/fl} *Lyz2-Cre* BMDMs treated LPS (1 μ g ml⁻¹) for 3 h, *E.coli* (MOI, 20) or *L. monocytogenes* (MOI, 10) for 1 h on different stiffness culture matrix of 5kPa, 35 kPa, or glass, followed by staining with MitoSOX or CellROX. (F) Immunoblot analysis of phosphorylated p-p38, p-Jnk and p-Mob1, Mob1, and γ -tubulin in *Piezo1*^{fl/fl} or *Piezo1*^{fl/fl} *Lyz2-Cre* BMDMs stimulated for 0, 10 or 30 min with LTA (1 μ g ml⁻¹).

REVIEWERS' COMMENTS

Reviewer #2 (Remarks to the Author):

The authors have added the control experiments I suggested for microscopy. I have no further concerns.

Point-by-point response to reviewers

TLR4 signalling via Piezo1 engages and enhances the macrophage mediated host response
during bacterial infection

(Manuscript # NCOMMS-20-23301C)

Reviewer #2 (Remarks to the Author):

The authors have added the control experiments I suggested for microscopy. I have no further concerns.

Answer: We thank the reviewer for the positive comments.